# Novel RNA viruses associated with *Plasmodium vivax* in human malaria and *Leucocytozoon* parasites in avian disease

Justine Charon[1], Matthew J. Grigg[2,3], John-Sebastian Eden[1,4], Kim A. Piera[2], Hafsa Rana[4], Timothy William[3,5,6], Karrie Rose[7], Miles P. Davenport[8], Nicholas M. Anstey[2,3], Edward C. Holmes[1]*

1 Marie Bashir Institute for Infectious Diseases and Biosecurity, Charles Perkins Centre, School of Life and Environmental Sciences and Sydney Medical School, The University of Sydney, Sydney, New South Wales, Australia, 2 Global and Tropical Health Division, Menzies School of Health Research and Charles Darwin University, Darwin, Northern Territory, Australia, 3 Infectious Disease Society Kota Kinabalu Sabah – Menzies School of Health Research Clinical Research Unit, Kota Kinabalu, Sabah, Malaysia, 4 Centre for Virus Research, Westmead Institute for Medical Research, Westmead, New South Wales, Australia, 5 Clinical Research Centre – Queen Elizabeth Hospital, Kota Kinabalu, Sabah, Malaysia, 6 Gleneagles Hospital, Kota Kinabalu, Sabah, Malaysia, 7 Australian Registry of Wildlife Health, Taronga Conservation Society Australia, Mosman, New South Wales, Australia, 8 Kirby Institute for Infection and Immunity, University of New South Wales, Sydney, New South Wales, Australia

* edward.holmes@sydney.edu.au

**Data Availability Statement:** The raw sequence data sets generated in this study, from which all human reads have been depleted, are available at the Sequence Read Archive database (BioProject PRJNA589654) at the following link - https://www.

## Abstract

Eukaryotes of the genus *Plasmodium* cause malaria, a parasitic disease responsible for substantial morbidity and mortality in humans. Yet, the nature and abundance of any viruses carried by these divergent eukaryotic parasites is unknown. We investigated the *Plasmodium* virome by performing a meta-transcriptomic analysis of blood samples taken from patients suffering from malaria and infected with *P. vivax*, *P. falciparum* or *P. knowlesi*. This resulted in the identification of a narnavirus-like sequence, encoding an RNA polymerase and restricted to *P. vivax* samples, as well as an associated viral segment of unknown function. These data, confirmed by PCR, are indicative of a novel RNA virus that we term Matryoshka RNA virus 1 (MaRNAV-1) to reflect its analogy to a "Russian doll": a virus, infecting a parasite, infecting an animal. Additional screening revealed that MaRNAV-1 was abundant in geographically diverse *P. vivax* derived from humans and mosquitoes, strongly supporting its association with this parasite, and not in any of the other *Plasmodium* samples analyzed here nor *Anopheles* mosquitoes in the absence of *Plasmodium*. Notably, related bi-segmented narnavirus-like sequences (MaRNAV-2) were retrieved from Australian birds infected with a *Leucocytozoon*—a genus of eukaryotic parasites that group with *Plasmodium* in the Apicomplexa subclass hematozoa. Together, these data support the establishment of two new phylogenetically divergent and genomically distinct viral species associated with protists, including the first virus likely infecting *Plasmodium* parasites. As well as broadening our understanding of the diversity and evolutionary history of the eukaryotic virosphere, the restriction to *P. vivax* may be of importance in understanding *P. vivax*-specific biology in humans and mosquitoes, and how viral co-infection might alter host responses at each stage of the *P. vivax* life-cycle.

ncbi.nlm.nih.gov/sra/PRJNA589654. The consensus MaRNAV-1 and MaRNAV-2 partial genome sequences and the corresponding translated ORFs are available at GenBank under the accessions MN698826-MN698831.

**Funding:** This work was supported by an Australian Research Council Australian Laureate Fellowship awarded to ECH (FL170100022), the Australian National Health and Medical Council (grants #1037304 and #1045156; Fellowships to NMA [#1042072] and MJG [#1138860]), and the Australian Centre of Research Excellence in Malaria Elimination. The Sabah malaria research program is supported by US National Institutes of Health (R01 AI116472-03) The funders had no role in study design, data collection and analysis, decision to publish, or preparation of the manuscript.

**Competing interests:** The authors have declared that no competing interests exist.

## Author summary

While parasites are a major cause of human disease, they can themselves be infected by viruses. We asked whether three of the major malaria-causing parasites in humans—*Plasmodium vivax*, *P. falciparum* and *P. knowlesi*—were also infected by viruses. To this end we performed total RNA-Sequencing ("meta-transcriptomics") on human blood samples infected with these *Plasmodium* species. This resulted in the discovery of an abundant bi-segmented virus—Matryoshka RNA virus 1 (MaRNAV-1)—in all *P. vivax* samples tested (but no other *Plasmodium* species) that contains a replicase segment related to those of narnaviruses, arguably the simplest type of RNA viruses discovered to date. By screening for MaRNAV-1 in a larger set of *Plasmodium* species we revealed a strong specificity between this virus and *P. vivax*, as well as the presence of a related virus—MaRNAV-2—in avian *Leucocytozoon* hematozoa parasites. This is the first discovery of a *Plasmodium*-associated virus and will assist in revealing the deep evolutionary history of RNA viruses and our understanding of *Plasmodium* biology and disease processes.

## Introduction

Viruses are the most abundant biological entities on Earth, replicating in diverse host organisms [1]. Although there has been an expansion of metagenomic studies dedicated to exploring this immense virosphere [2–6], our knowledge of the viral universe remains limited, with only a minute fraction of eukaryotic species sampled to date [7]. This knowledge gap is especially wide in the case of unicellular eukaryotes (i.e. protists), including those responsible for parasitic disease in humans, on which only a small number of studies have been performed.

Viral-like particles in parasites were first observed by electron microscopy as early as the 1960's in various protozoa from the apicomplexan and kinetoplastid phyla [8]. The first molecular evidence for the existence of protozoan viruses was obtained in the late 1980s, resulting in the characterization of double-strand (ds) RNA viruses in the human parasites *Giardia*, *Leishmania*, *Trichomonas* and *Cryptosporidium* [9–14]. More recently, single-stranded narnavirus-like and bunyavirus-like RNA viruses were identified in trypanosomatid parasites, including *Leptomonas seymouri*, *Leptomonas moramango*, *Leptomonas pyrrhocoris* and *Crithidia* sp. [15–18]. However, our knowledge of protozoan viruses is clearly limited, with many of those identified stemming from fortuitous discovery.

The identification and study of protozoan viruses is also important for our understanding of so-called "Russian doll" ("Matryoshka" in Russian) infections [19], in which parasites are themselves infected by other microbes. A key question is whether viruses of parasites can in turn have an impact on aspects of parasite pathogenesis? An increasing number of studies have demonstrated that dsRNA viruses of protozoa can affect key aspects of parasite biology, including their virulence, in a variety of ways [20]. For instance, data from *Leishmania guyanensis* and *Trichomonas vaginalis* strongly suggest a link between parasite pathogenesis and the presence of Leishmania RNA virus 1 (LRV1) and Trichomonas vaginalis virus, respectively [21–23]. By increasing the inflammatory response in the host these viruses could in theory enhance human pathogenesis [24–25]. Interestingly, associations have also been observed between LRV1-infected *L. guyanensis* or *L. braziliensis* and treatment failure in patients with leishmaniasis [26–27].

Viral co-infection also has the potential to alter protozoan biology and/or attenuate the mammalian host response, leading to greater replication or persistent protozoan infection, in turn promoting ongoing parasite transmission. Persistence (i.e. avirulent infection) has been

proposed in the case of Cryptosporidium parvum virus 1 (CSpV1) that infects the apicomplexan *Cryptosporidium* [28], and increased *C. parvum* fecundity has been demonstrated in isolates experiencing viral co-infection [29]. Viral infections may also have a deleterious effect on parasite biology, adversely impacting such traits as growth and adhesion in the case of axenic cultures of *Giardia lamblia* [10]. Clearly, the effects and underlying molecular basis of any consequences that protozoal viruses have on their hosts, including in the context of pathogenesis, requires rigorous investigation. Documenting novel protozoal viruses is an obvious first step in this process.

Remarkably, nothing is known about those viruses that infect species of *Plasmodium* (order Haemosporida)—obligate apicomplexan parasites of vertebrates and insects. In vertebrate hosts, these protozoa first infect the liver cells as sporozoites where they mature into schizonts. Resulting merozoites are then released into the bloodstream to undergo asexual multiplication in red blood cells. A portion of these replicating asexual forms can differentiate into gametocytes which, following ingestion by blood-feeding female *Anopheles* mosquitoes, develop into sporozoites and are transmitted to another host via mosquito saliva. The genus *Plasmodium* currently comprises approximately 100 species that infect various mammals, birds and reptile hosts. Among these, six species commonly infect humans and are important causative agents of human malaria: *P. falciparum*, *P. vivax*, *P. malariae*, *P. ovale curtisi*, *P. ovale wallikeri*, and *P. knowlesi*. Despite an early observation of viral-like particles in cytoplasmic vacuoles of simian *P. cynomolgi* sporozoites [30], no viruses have been discovered in the parasites responsible for malaria.

With 219 million cases reported in 2017 in 90 countries around the world, malaria continues to be the most important protozoan disease affecting humans [31]. Despite ongoing and considerable global public health efforts, recent progress in reducing the disease burden due to malaria has stalled. Reasons include the emergence of resistance to insecticides in the mosquito vectors and parasite resistance to antimalarial drugs in humans. In addition, the large number of asymptomatic and/or submicroscopic *Plasmodium* infections in peripheral blood are an important source of transmission and pose a major challenge to control and eradication strategies [32]. This is compounded by the ability of some *Plasmodium sp.*, including *P. vivax*, *P. ovale curtisi*, and *P. ovale wallikeri*, to form latent liver stages and later relapse. They also illustrate the need for approaches targeting the human parasite reservoir rather than treating only those with clinical disease.

There is an obvious interest in identifying viruses associated with human *Plasmodium* species from both an evolutionary and clinical perspective. The presence of RNA viruses infecting hematozoa parasites have been largely overlooked, although their position deep in the eukaryotic phylogeny means that they may constitute a valuable source of information to help understand early events in the evolution of eukaryotic RNA viruses. Knowledge of *Plasmodium*-specific viral infection may also provide insights into parasite biology in humans and mosquitoes, with the potential for identifying preventative or therapeutic strategies.

## Results

### *Plasmodium*-infected human samples

To investigate the virome of *Plasmodium* parasites that infect and cause disease in humans, we performed a meta-transcriptomic study of three species—*P. vivax* (hereby denoted Pv), *P. knowlesi* (Pk) and *P. falciparum* (Pf). These samples were obtained from 7, 6 and 5 malaria patients, respectively, at different locations in the state of Sabah, east Malaysia (S1 Table) [33]. All patients with malaria had uncomplicated disease. An additional library of 6 uninfected patients was also included as a negative control. All infected blood samples were validated for

their corresponding *Plasmodium* species (S1 Table). Microscopic parasite counts from peripheral blood films revealed similar densities (i.e. no significant differences, p-value = 0.7) between the three *Plasmodium* species, with parasitemia centered around 6000–8000 parasites per µL (S1 Fig).

## Sample processing

Homogenous and equimolar ratios of each of the total RNA samples were used to prepare RNA-Seq libraries. Sequencing depth was similar for all samples, with 17±0.5 million reads obtained (Fig 1A). The human and ribosomal RNA (rRNA) read depletion drastically reduced the number of reads in both the non-infected and Pf data sets (93 and 81% of reads filtered, respectively) (Fig 1B) and to a lesser extent in Pk and Pv (only 42–57% of reads removed). Pf

**(A)**

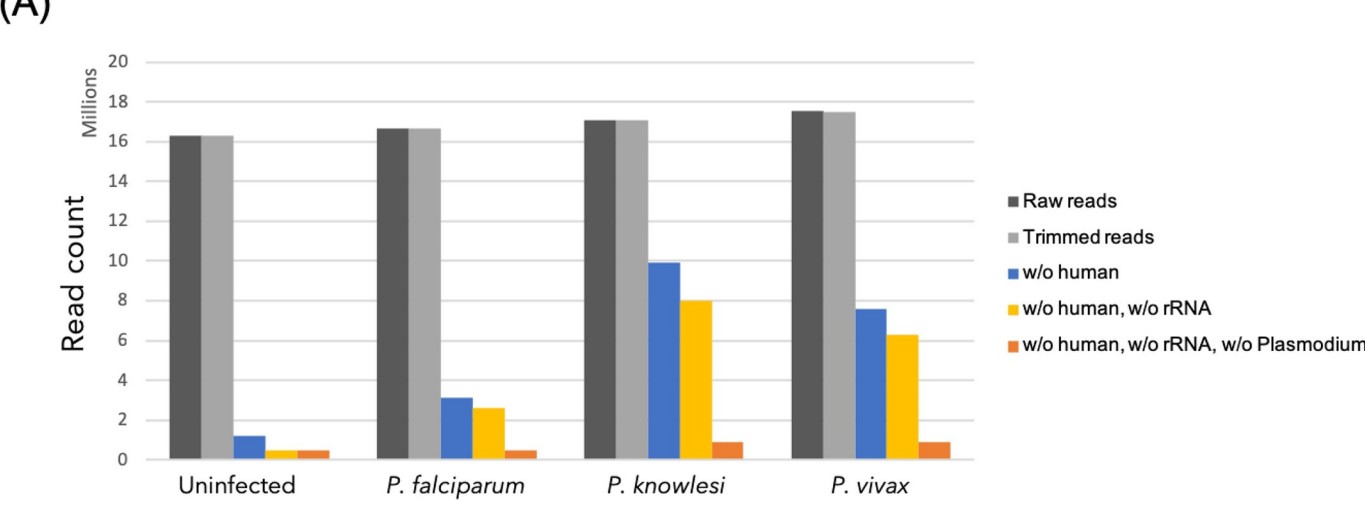

**(B)**

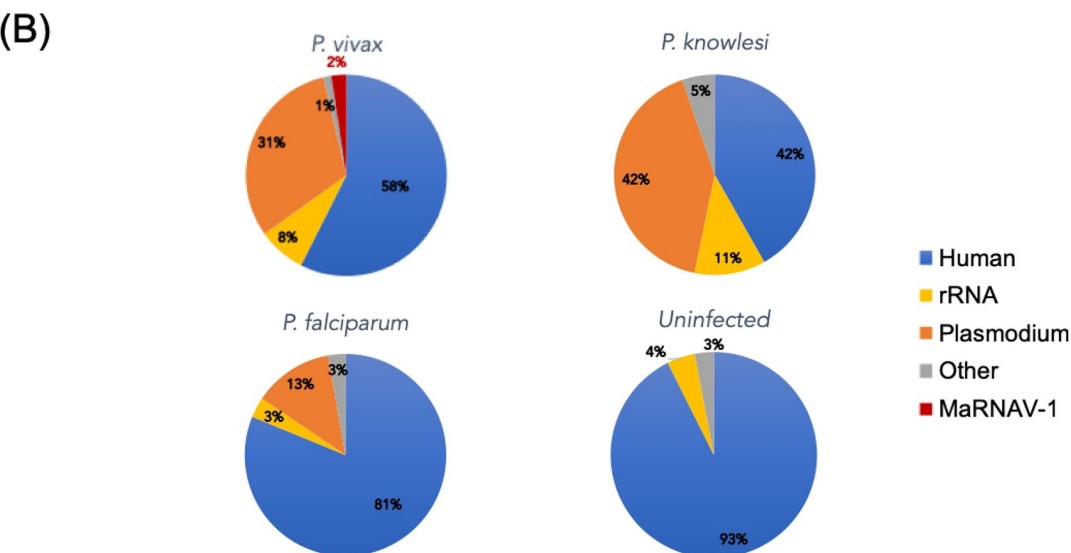

**Fig 1. Host read depletion in RNA-Seq libraries.** Reads were mapped against rRNA SILVAdb (SortMeRNA tool), the human genome, and the genomes of three *Plasmodium* species. (A) Efficiency of successive read filtering (rRNA and host sorting). (B) Proportion of major host transcripts in each data set. The number of reads mapping to the human genome, *Plasmodium* sp. genomes and MaRNAV-1 are expressed as the percentage of trimmed reads for each library.

transcripts were less abundant in libraries than those in Pk and Pv. Finally, the contig assemblies performed on each library depleted for rRNA, human and *Plasmodium* reads were almost equally successful for all libraries, with a similar contig length distribution between the data sets (S2 Fig).

## Virus discovery in *Plasmodium vivax*-infected blood samples

**Discovery of a bi-segmented RNA Narna-like virus.** Ribosomal RNA-depleted data sets were submitted to BLASTx against a database containing all the RNA-dependent RNA polymerase (RdRp) protein sequences available at the NCBI. We focused on this protein as it is the mostly highly conserved among RNA viruses and hence constitutes the best marker for detecting their presence and performing expansive phylogenetic analyses. False-positive hits (i.e. non-viral contigs) were discarded by using a second round BLAST against the non-redundant protein (nr) database and removing contigs with non-viral top hits. Notably, true-positive RdRp signals were only found in the Pv library (Table 1).

These contigs all correspond to variants of the same gene from the Trinity assemblies, and all share their highest sequence similarity score (between 42.6 and 42.9% identity, Table 1) with the RdRp of Wilkie narna-like virus 1, an unclassified virus related to the narnaviruses recently identified in mosquito samples [34]. The mapping of Pv-reads against this newly-described viral-like genome revealed that the virus-like contig was highly abundant in the Pv library, comprising approximately 1.6% of all reads (from which rRNA has been excluded) (Fig 1). A more detailed characterization of this virus is presented below. To detect homologs to this newly identified RNA-virus-like contig in the other *Plasmodium* species, DN5867 contigs were used as the reference for another round of BLASTx: this analysis revealed no matches in either the Pf or Pk data sets.

The apparent bias in virus composition between libraries likely reflects differences in their virome composition rather than experimental bias, since the quality of samples, the depth of sequencing, and the contig assembly were similar among the four libraries. However, it is also possible that it in part reflects the limits of BLASTn/BLASTx sequence-based homology detection methods to identify highly divergent RNA viruses. To help overcome this limitation, and try to identify any highly divergent RNA viruses, we performed a BLASTn/BLASTx search on the contigs using the nt and nr databases, respectively. Assuming that RNA virus-like genomic sequence would be of a minimum length set to 1000 nt (as there are no RNA viruses shorter than this), and to make the analysis computationally tractable, only the "orphan" contigs (i.e. those without any match in any of the nt or nr databases) longer than 1000nt were used for further analysis (S3 Fig). To identify remote virus signal from these sequences, a second round of BLASTx search was conducted with lower levels of stringency: this revealed no clear hits to RNA viruses.

**Table 1. Results of the RdRp BLASTx analysis.**

| Contig query | Length | estimated_count | BLASTn | BLASTx best hit | %ID | e-value | taxID | Virus |
|---|---|---|---|---|---|---|---|---|
| Pv_1_DN5867_c0_g1_i1 | 2924 | 234605.7 | No hit | YP_009388589.1 RdRp | 42.8 | 1.30E-170 | 2010280 | Wilkie narna-like virus 1 |
| Pv_1_DN5867_c0_g1_i2 | 3023 | 77610.78 | No hit | YP_009388589.1 RdRp | 42.8 | 1.10E-170 | 2010280 | Wilkie narna-like virus 1 |
| Pv_1_DN5867_c0_g1_i3 | 3023 | 286828.4 | No hit | YP_009388589.1 RdRp | 42.6 | 3.70E-184 | 2010280 | Wilkie narna-like virus 1 |
| Pv_1_DN5867_c0_g1_i4 | 2141 | 105799.2 | No hit | YP_009388589.1 RdRp | 42.9 | 1.50E-185 | 2010280 | Wilkie narna-like virus 1 |
| Pv_1_DN5867_c0_g1_i5 | 3023 | 1834.39 | No hit | YP_009388589.1 RdRp | 42.8 | 1.10E-170 | 2010280 | Wilkie narna-like virus 1 |
| Pv_1_DN5867_c0_g1_i6 | 2045 | 79319.65 | No hit | YP_009388589.1 RdRp | 42.8 | 6.60E-171 | 2010280 | Wilkie narna-like virus 1 |
| Pv_1_DN5867_c0_g1_i7 | 1496 | 13751.44 | No hit | YP_009388589.1 RdRp | 42.8 | 8.30E-171 | 2010280 | Wilkie narna-like virus 1 |

Finally, we employed a structural-homology based approach to virus discovery, utilizing information on protein 3D structure rather than primary amino acid sequence, as the former can be safely assumed to be more conserved than the latter [35] and is therefore predicted to be better able to reveal distant evolutionary relationships. Accordingly, hypothetical ORFs were predicted from orphan contigs and Hidden Markov model (HMM) searches combined with 3D-structure modelling were performed on the corresponding amino acid sequences using the Phyre2 web portal [36]. Again, this revealed no reliable signal for the presence of highly divergent viruses in the RNA sequences obtained here.

Notably, with 122,452 reads mapped to it, the Pv unknown contig retained is highly expressed, possessing a similar level of abundance as the newly-identified Pv RdRp-like contig. Specifically, the abundance of these two contigs were in the same range, with Reads per Kilobase per Million (RPKM) of the Pv RdRp-like and Pv unknown contigs of 2.9 and 2.6, respectively. Such a similarity in abundance levels supports the existence of a bi-segmented RNA virus. Finally, the 3kb RdRp-segment described in our *P. vivax* samples is also within the range of the genome lengths seen in other members of the *Narnaviridae* (2.3 to 3.6 kb).

**Narna-like virus genome and protein annotation.** The two segments of our putative narnavirus were both validated by Reverse Transcriptase-PCR (RT-PCR) in each of the seven *P. vivax* samples used for this study, but were not found in the *P. knowlesi*, *P.falciparum* nor the uninfected samples (Fig 2). Corresponding amplicons were then Sanger-sequenced to define both the inter-sample sequence diversity. We named this new virus Matryoshka RNA virus 1 (MaRNAV-1) because of its "Russian doll" composition, reflecting a virus that infects a parasite (protist) that infects an animal. The Sanger sequencing results of MaRNAV-1 for each Pv sample revealed a very high level of sequence conservation in the RdRp-encoding segment ("RdRp-segment"; Fig 3A, left).

Virus ORFs were predicted using the ORFinder NCBI tool and corresponding amino acid sequences were obtained and aligned (Fig 3A, right). This revealed that the nucleotide polymorphisms described above were also present at the amino acid level, even though these sequences were still highly conserved (98–100%), especially in the RdRp. An additional attempt at functional annotation was performed but did not reveal any additional functional motifs or domains aside from the RdRp.

The second segment, the presence of which distinguishes MaRNAV-1 virus from all other narnaviruses, was also highly conserved between *P. vivax* samples (between 95 and 100% identity at the nucleotide level) and is likely to encode two protein products of 205 and 163 amino acids in length through two overlapping ORFs (Fig 3B). Unfortunately, the level of sequence divergence between this second segment and all other sequences available at NCBI meant that no functional annotations were possible.

Very few nucleotide polymorphisms were observed between the viruses identified from samples #1, #3, #4, #5, #6 and #10, which effectively showed 100% sequence identity (Fig 3C). In contrast, the MaRNAV-1 segments I and II from sample #2 are more distant, with 93% and 95% nucleotide identity, respectively, to the other sequences (Fig 3C) and 98% and 96% (ORF1) or 97% (ORF2) identity, respectively, at the amino acid level. Why sample #2 exhibits more diversity is unclear. It is similarly uncertain why segments I and II differ in levels of genetic diversity. Obtaining a satisfactory answer to this question will require more detailed information on protein function.

**Phylogenetic analysis of MaRNAV-1.** To link the newly-identified *Plasmodium vivax* virus to the known diversity of RNA viruses, we performed a phylogenetic analysis with the sequence newly acquired here and the closest available relatives identified with BLASTx (Fig 4). To our knowledge, this is the first description of a virus associated with *Plasmodium* spp. and few apicomplexan-related viruses have been isolated to date. Hence, it is not

### P. vivax

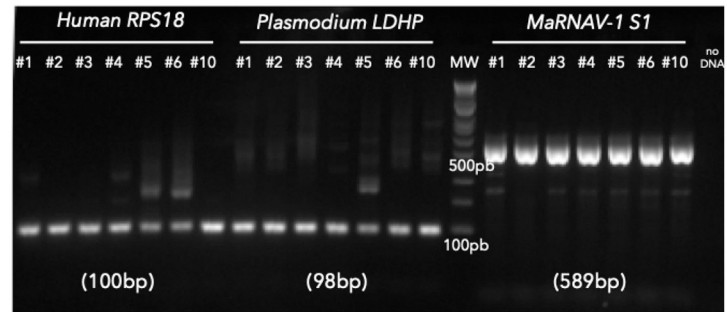 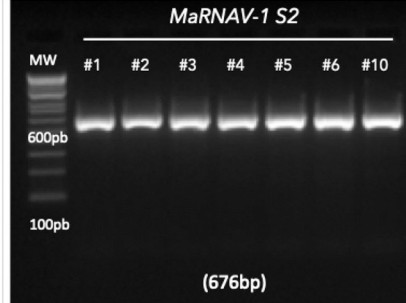

### P. knowlesi

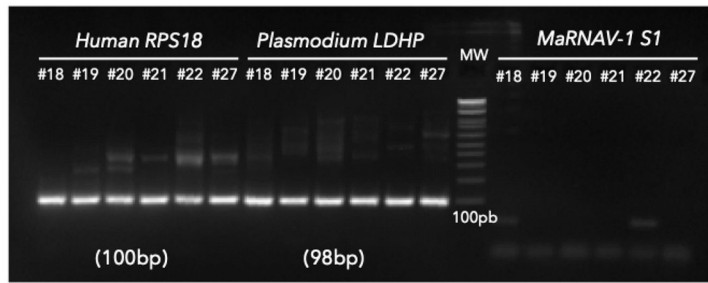 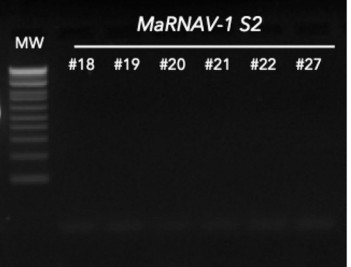

### P. falciparum

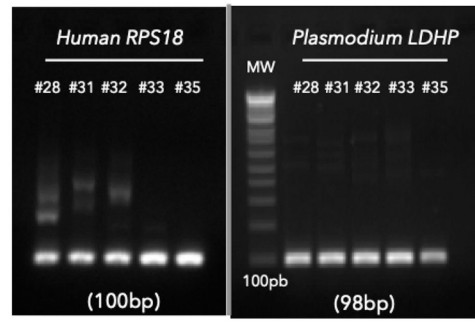 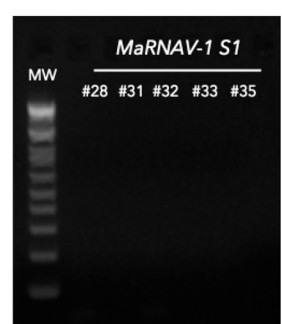 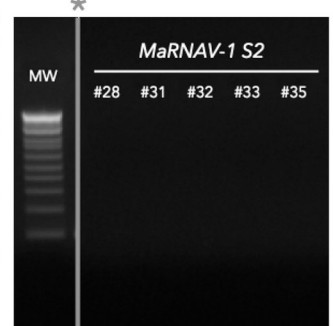

### Non-infected samples

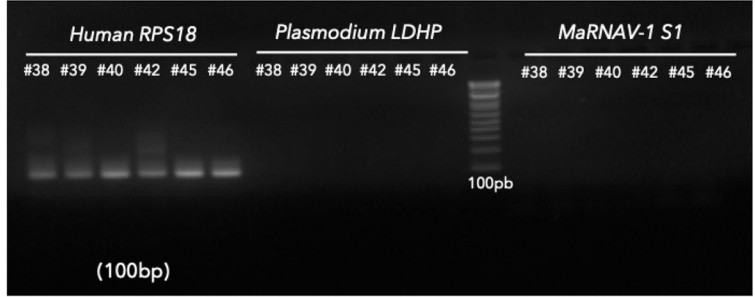 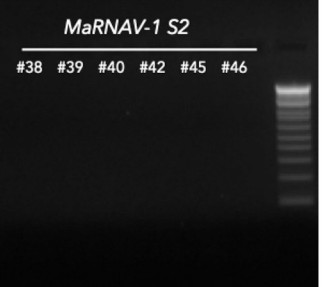

**Fig 2. RT-PCR confirmation of host and virus-like sequences in all *Plasmodium*-infected and non-infected samples used in this study.** From left to right: RT-PCR of each of the samples using human RPS18 primers, *Plasmodium* LDHP primers, MaRNAV-1 Segment I (S1) primers and MaRNAV-1 Segment II (S2) primers. Numbers in parentheses correspond to the expected size of the corresponding amplicon. (*) indicate different parts of the same gel that have been cropped for ease of visualization only.

(A)

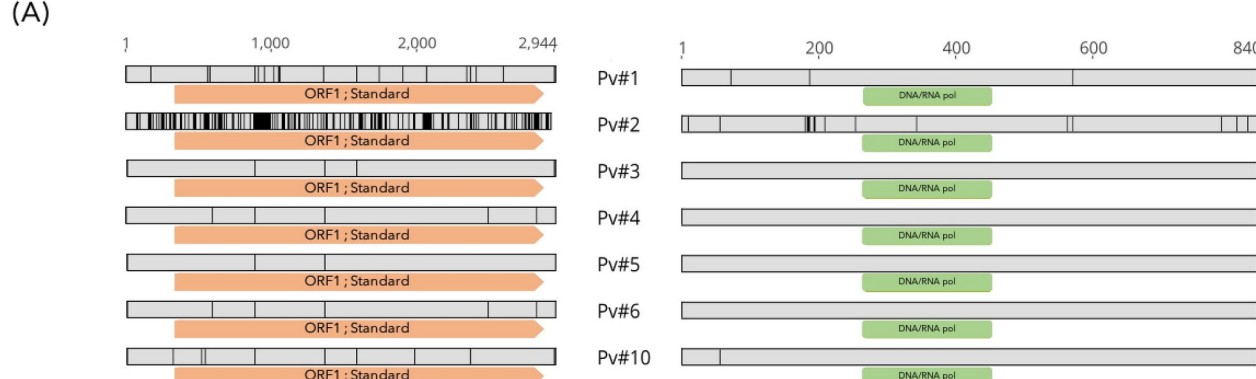

(B)

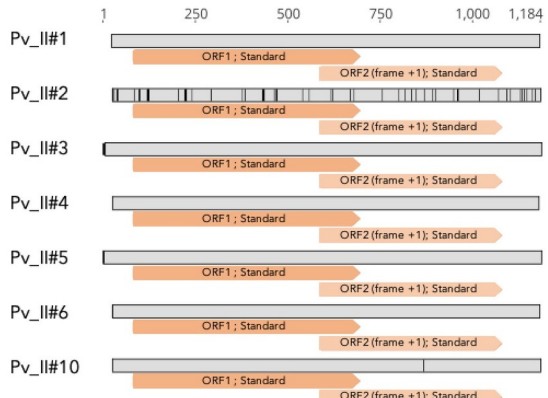

(C)

| Segment I | #1 | #2 | #3 | #4 | #5 | #6 | #10 |
|---|---|---|---|---|---|---|---|
| #1 | - | 93 | 99 | 99 | 99 | 99 | 99 |
| #2 | 93 | - | 93 | 93 | 93 | 93 | 93 |
| #3 | 99 | 93 | - | 100 | 100 | 100 | 100 |
| #4 | 99 | 93 | 100 | - | 100 | 100 | 100 |
| #5 | 99 | 93 | 100 | 100 | - | 100 | 100 |
| #6 | 99 | 93 | 100 | 100 | 100 | - | 100 |
| #10 | 99 | 93 | 100 | 100 | 100 | 100 | - |

| Segment II | #1 | #2 | #3 | #4 | #5 | #6 | #10 |
|---|---|---|---|---|---|---|---|
| #1 | - | 95 | 100 | 100 | 100 | 100 | 100 |
| #2 | 95 | - | 95 | 95 | 95 | 95 | 95 |
| #3 | 100 | 95 | - | 100 | 100 | 100 | 100 |
| #4 | 100 | 95 | 100 | - | 100 | 100 | 100 |
| #5 | 100 | 95 | 100 | 100 | - | 100 | 100 |
| #6 | 100 | 95 | 100 | 100 | 100 | - | 100 |
| #10 | 100 | 95 | 100 | 100 | 100 | 100 | - |

**Fig 3. Genomic organization and sequence polymorphism of MaRNAV-1 in the seven *P. vivax*-infected blood samples.** (A) RdRp-segment analysis. Left: Nucleotide sequence alignment and ORF prediction (orange boxes). Right: Protein sequence alignment and InterPro domains predicted (green boxes). (B) Nucleotide polymorphism and ORFs predicted from the segment II in *P. vivax samples*. Sequence polymorphisms are highlighted in black. (C) Distance matrix of segment I (left) and segment II (right) with percentage of identity obtained at the nucleotide level.

surprising that only low levels of amino acid sequence similarity (between 15 and 54%) were found in comparisons between MaRNAV-1 and the closest related RNA viruses available at NCBI. Importantly, however, the most closely related viruses were consistently classified as members of the family *Narnaviridae* (genus *Narnavirus*)—a group of single-stranded positive-sense RNA viruses known to infect such host species as fungi, plants and, importantly, protists (Fig 4). The most closely related virus—Wilkie narna-like virus 1—was recently identified in a

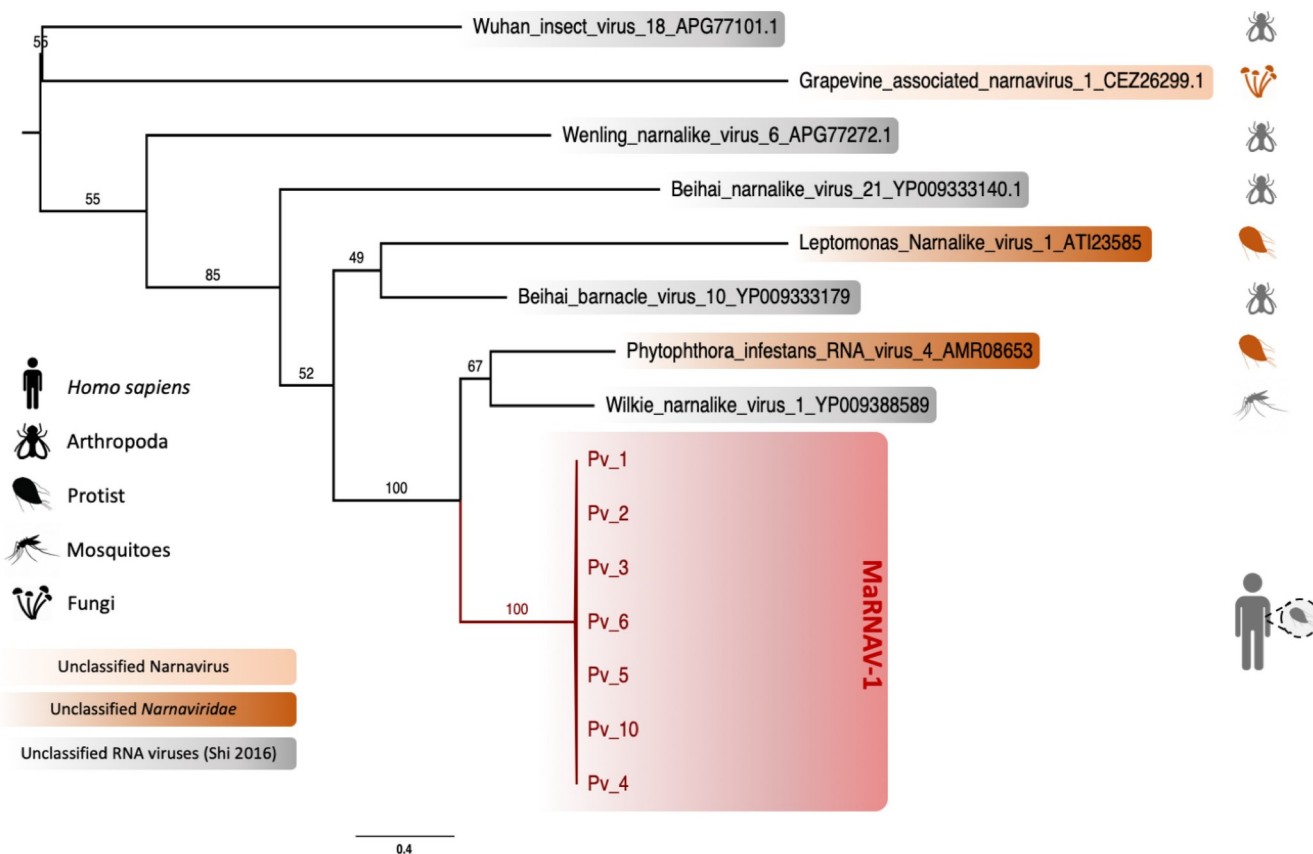

**Fig 4. Phylogenetic analysis of MaRNAV-1 associated with *Plasmodium vivax*.** Boxes refer to the newly-described MaRNAV-1 viral sequences obtained in this study (red) or to RNA viruses classified as members of the *Narnaviridae* (dark orange), genus *Narnavirus* (light orange) or currently unclassified (grey). Taxa corresponding to the validated (coloured icons, right) and non-validated (grey icons, right) hosts are reported on the left part of the tree. Bootstrap values are indicated on each branch. The tree is mid-point rooted for clarity only.

large-scale survey of mosquitoes [34] and is yet to be formally taxonomically assigned. Although the low abundance of this virus meant that no host could be conclusively assigned, a preliminary study suggested that it was unlikely to be a virus of mosquitoes, such that it could, in theory, infect a protozoan within the mosquitoes. In addition, two of the other narnaviruses most closely related to MaRNAV-1 virus—*Leptomonas seymouri narna-like virus 1* (LepseyNLV1) and *Phytophtora infestans RNA virus 4* (PiRV-4)—are seemingly associated with unicellular eukaryotes—*Leptomonas seymouri* and *Phytophtora infestans*, respectively [17, 37].

The second putative segment found in all the *P. vivax* samples described here also aligned with the second segment present in *LepseyNLV1* which similarly encodes two overlapping ORFs (KU935605.1), even though they share little sequence similarity (only 14–18% identity at the amino acid level for ORF1 and ORF2, respectively). This high divergence explains why this sequence was not identified in previous BLASTx analyses and precluded more detailed phylogenetic analysis.

## Virus-host assignment

A major challenge for all metagenomic studies is accurately assigning viruses to hosts as they could in reality be associated with host diet, the environment surrounding the sampling site, or a co-infecting micro-organism. In tentatively assigning hosts we assumed that: (i) a virus

with a high abundance is likely to be infecting a host found also in high abundance, (ii) a virus consistently found in association with one particular host is likely to infect that host, (iii) a virus that is phylogenetically related to those previously identified as infecting a particular host taxa is likely to infect a similar range of host taxa, and (iv) a virus and a host that share identical genetic code and/or similar codon usage or dinucleotide compositions are likely to have adapted and co-evolved, indicative of a host-parasite interaction.

**Eukaryotic host read profiling.** To initially assess whether MaRNAV-1 is likely to infect *Plasmodium* rather than other intra-host microbes and co-infecting parasites, the host taxonomy of the BLASTn/x top hits for each contig of the human and *Plasmodium*-depleted *P. vivax* library were compared to their respective size and abundance. However, this analysis revealed only a small number of short contigs associated with fungi and bacteria (S4 Fig). This result is of note as the usual hosts associated with narnaviruses are fungi, and the closest relatives have been found in mosquito samples, although the true host of this virus could in theory be protozoal. Among the Metazoa identified, all the contigs were associated with vertebrates, rather than members of the Arthropoda or Nematoda. Hence, *Plasmodium vivax* appears to be the most likely host of the newly-identified MaRNAV-1 virus.

**Comparison of codon usage and dinucleotide composition.** Host adaptation can result in specific patterns of codon and dinucleotide usage. We compared the codon usage observed in MaRNAV-1 to those of the potential host organisms. The Codon Adaptation Index (CAI) measures the similarity in synonymous codon usage between a gene and a reference set and was assessed for MaRNAV-1 using *H. sapiens*, *P. vivax* and *Anopheles* genera mosquitoes (pool of 7 species) as reference data sets. However, as none of the CAI/eCAI values obtained were significant ($<1$) (S5 Fig), no conclusions could be drawn regarding the potential host of MaRNAV-1. In a complementary approach, we compared the dinucleotide composition bias between the newly identified virus and the potential hosts [38]. Again, the dinucleotide frequencies in the two potential hosts *An. gambiae* and *P. vivax* revealed strong similarities that prevented us from identifying any signature of viral adaptation (S6 Fig).

**Investigation of the MaRNAV-1 and *Plasmodium* sp. association using the Sequence Read Archive (SRA).** To further test for an association between MaRNAV-1 and *Plasmodium* parasites, we performed a wider investigation of the occurrence of this virus in *Plasmodium*-infected samples and other *Plasmodium* species for which RNA-Seq data were available on the SRA. These data sets comprised *P. chabaudi*, *P. cynomolgi*, *P. falciparum*, *P. yoelli*, *P. knowlesi* and *P. berghei* (the relevant Bioprojects are listed in S2 Table). Reads counts $<10$ were considered too low to be informative.

In total, 1682 RNA-Seq data sets from *Plasmodium*-related projects on the SRA were screened for the presence of MaRNAV-1 using BLASTx. Reads mapping to MaRNAV-1 were identified in 45 libraries (including biological replicates), all of which belonged to *P. vivax* (Fig 5). Among the *P. vivax*-related runs (S3 Table), none of the 31 uninfected or *P. falciparum*-infected samples contained MaRNAV-1 (Chi-squared test, p-value = 0, S7A Fig). This pattern is strongly suggestive of a specific association between MaRNAV-1 and *P. vivax*, rather than the result of experimental bias or contamination introduced during RNA extraction or sequencing. Moreover, MaRNAV-1 was found in 43% (13 out of 30) of the *P. vivax*-infected SRA samples investigated here (biological replicates omitted). No obvious biological or experimental features were identified that could reasonable explain these patterns of prevalence.

In addition, the detection of MaRNAV-1 in the SRA-based studies was independent of the geographic location of where the *P. vivax* isolates were sampled (Colombia, Cambodia and Thailand) and the sample type obtained (human blood, mosquito dissected salivary glands or *ex vivo* cultures) (Chi-squared tests, p-values $> 0.05$, S7C and S7D Fig). Hence, together, these results strongly support a specific association between MaRNAV-1 and *P. vivax*.

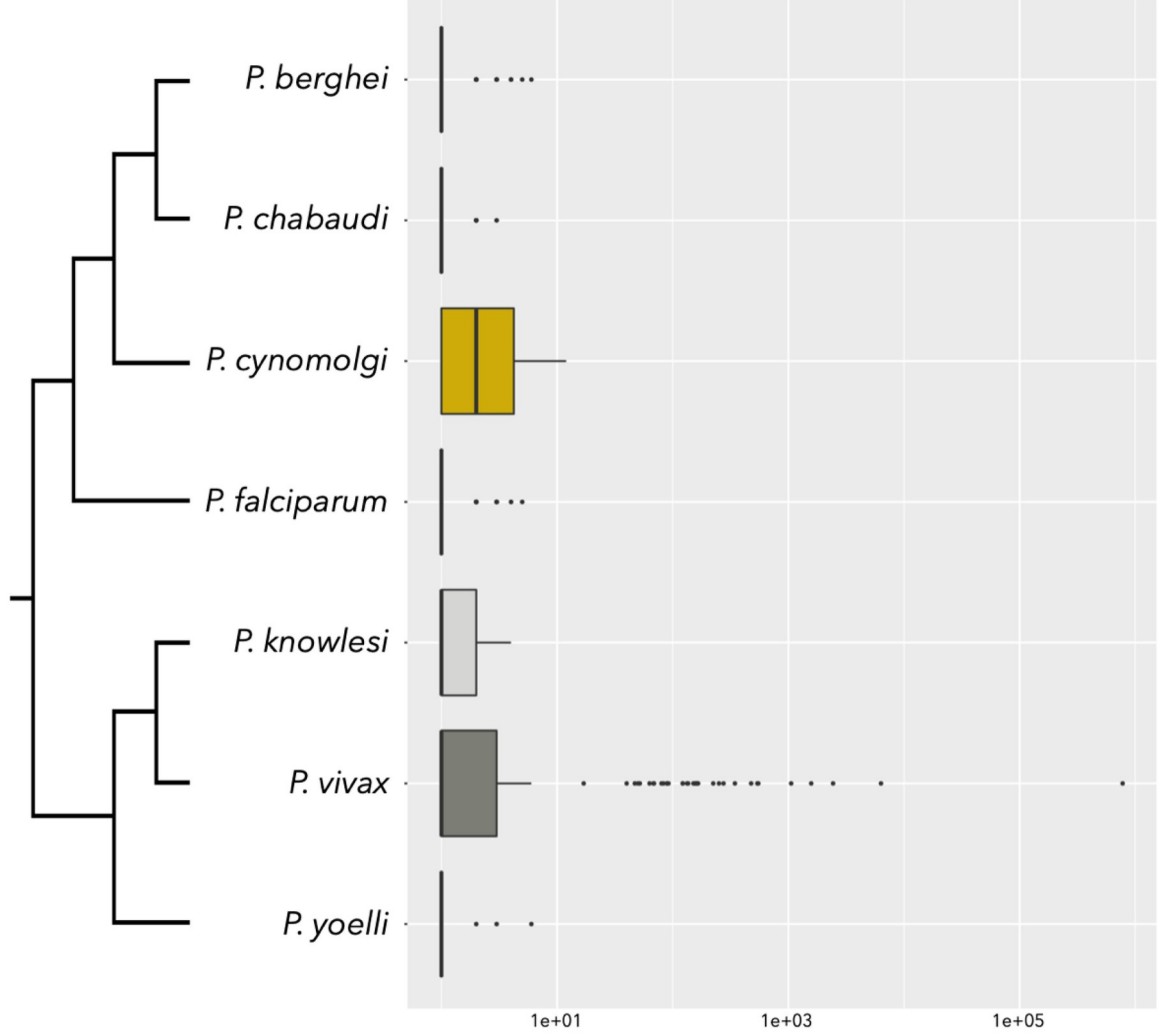

**Fig 5. Number of *Plasmodium* SRA reads aligning with the MaRNAV-1 sequence (RdRp-segment) using BLASTx (cut-off 1e-5).**

Finally, we performed an additional screen of the MaRNAV-1 on 42 SRA libraries from *P. vivax*-free mosquitoes belonging to seven Anopheles species that are considered among the major vectors of *P. vivax* in the Asian Pacific region (S4 Table) [39]. Notably, no MaRNAV-1-like reads were identified in these data, strongly supporting the association of MaRNAV-1 with the presence of *P. vivax* and not the mosquito. These results are also consistent with the observation that all the three *Plasmodium* species tested in the study site of Sabah share the same predominant mosquito vector—*A. balabacensis*—yet MaRNAV-1 sequences were only found in *P. vivax* samples.

## Analysis of SRA-derived MaRNAV-1 virus-like genomes

Narnavirus positive *P. vivax* data sets were further analyzed following the same workflow as described above. Hence, contigs were *de novo* assembled and re-submitted to BLASTx to extract full-length contigs corresponding to MaRNAV-1. The genomes obtained were

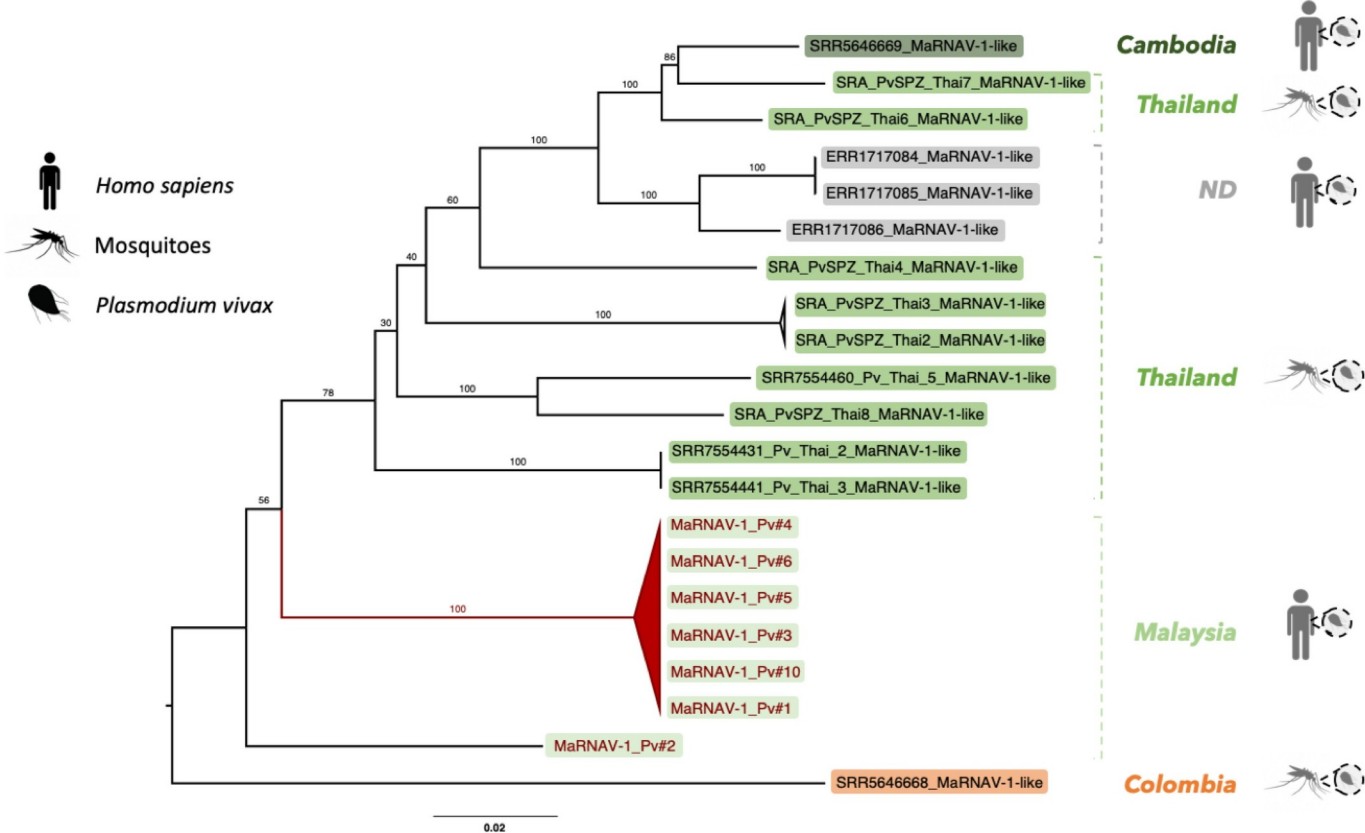

**Fig 6. Phylogenetic analysis, based on the RdRp, of the MaRNAV-1 documented here and from the *P. vivax* sequences available on the SRA.** Those viruses obtained in this study are shown in red while those from the SRA are shown in black. Sampling location and host characteristics (i.e. human-infected or mosquito-infected samples) are indicated on the right. Colored boxes indicate the samples collected in Asia (green), in South America (orange) or from unknown location (grey; ND: non-determined). The tree is mid-point rooted for clarity only.

validated and quantified using read mapping and overlapping contigs were merged to obtain full-length viral genomes.

A phylogenetic analysis of these sequences containing MaRNAV-1 was performed at the nucleotide level (Fig 6). Importantly, phylogenetic position was strongly associated with sampling location rather than the nature of the samples (i.e. human blood or *Anopheles sp.* mosquito salivary glands). This again reinforces the idea that these sequences come from a RNA virus infecting *Plasmodium sp.* rather than human or *Anopheles sp.* hosts. Despite this geographical association, all these newly identified RdRp-encoding sequences shared a high level of sequence nucleotide identity (85–100%). Promisingly, the sequence of the second segment identified in this study is also found in *P. vivax* SRA data sets and is strongly associated ($R^2 > 0.95$) with the presence of the RdRP-encoding segment (S8 Fig).

## Detection of MaRNAV-2 in *Leucocytozoon* parasites infecting birds

To investigate whether these narna-like sequences might infect a wider taxonomic distribution of hosts, we performed a complementary analysis of bird samples infected by members of the genus *Leucocytozoon*: apicomplexan parasites that belong to the same hematozoa subclass (of the Apicomplexa) as *Plasmodium*. These complementary studies were conducted on available RNA-Seq data previously obtained from liver, brain, heart and kidney tissues from Australian

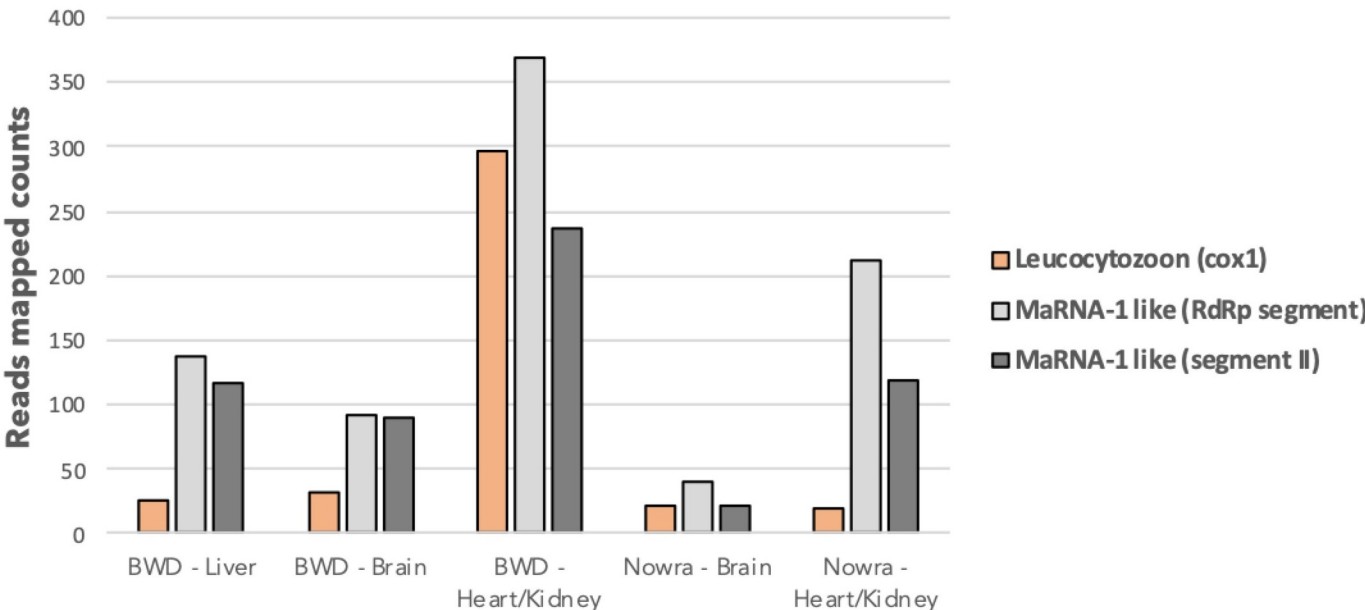

**Fig 7. Comparative abundance of *Leucocytozoon* and MaRNAV-2 transcripts in *Leucocytozoon*-infected avian RNA-Seq libraries.** Read counts from each RNA-seq data set that mapped to the *Leucocytozoon* mitochondrial Cox1 gene (light orange), MaRNAV-2 RdRp-like segment (light grey), and MaRNAV-2 Segment II (dark grey) sequences, determined using Bowtie2.

Magpie, Pied currawong and Raven birds collected at various time points in New South Wales, Australia (S5 Table). Using the newly-discovered MaRNAV-1 viral segments as references, a BLASTx analysis was performed on RNA-Seq data previously obtained for these samples. A first segment encoding a single predicted ORF of 859 amino acid long and containing the RdRp domain motif was retrieved and compared to the *P. vivax* MaRNAV-1 sequences (S9A Fig). A relatively high level of sequence similarity (73% identity at the amino acid level) was observed between the *Leucocytozoon*-infected birds and the viral sequences found in the *P. vivax* infected-humans. A second segment was also extracted from these avian libraries that exhibited strong similarities in terms of length, genome organization and sequence identity with the prediction of two overlapping ORFs, denoted ORF1 and ORF2, that encode proteins of 246 and 198 amino acids, respectively, and that share 48–52% amino acid identity with the MaRNAV-1 segment II ORFs (S9B Fig). The relative abundance of the *Leucocytozoon* and both segments of MaRNAV-1 like transcripts were assessed in all the five RNA-Seq libraries by counting the total reads that mapped to the respective sequences using Bowtie2 and showed an overall positive correlation ($R^2 = 0.75$), even though discrepancies can be observed between the libraries (Fig 7).

Next, we explored the association between the presence of the *Leucocytozoon* parasites and the MaRNAV-1 virus homologs by performing RT-PCR on each individual sample previously used to perform RNA-Seq. The two viral segments were always found as both-present or both-absent for all of the 12 avian samples analyzed (S5 Table, S10 Fig). In addition, in the majority of samples (25 of 27), the presence of the viral segments was directly linked to the presence of the parasite: that is, the virus was present only when the parasite was detected and absent in parasite-free samples (S5 Table). This supports the idea that the viral sequences screened are infecting the *Leucocytozoon* parasite rather than the bird carrying it. Because of its similarity to *P. vivax* MaRNAV-1, we term this *Leucocytozoon* parasite MaRNAV-2.

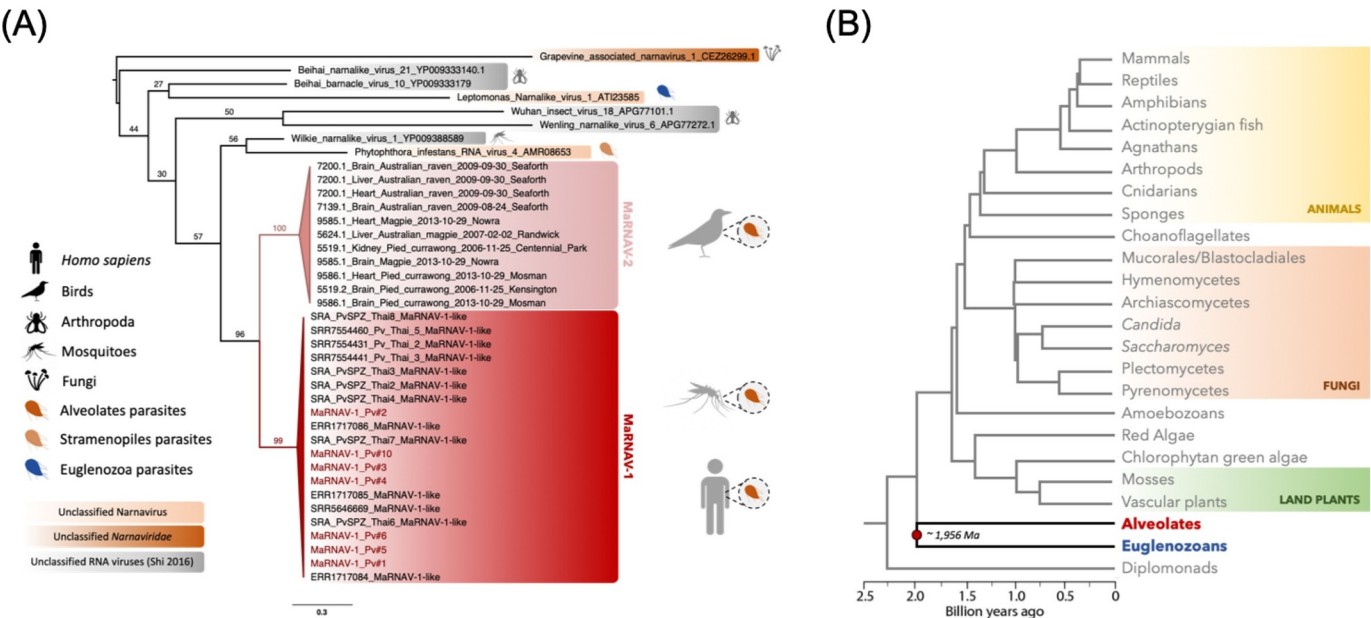

**Fig 8. Evolutionary relationships of the newly-identified hematozoa viral sequences (MaRNAV-1 and MaRNAV-2).** (A) Phylogeny of all the newly-identified viral sequences. Red box: *P. vivax* viruses MaRNAV-1 (human or mosquitoes infection stage). Pink box: *Leucocytozoon sp.* MaRNAV-2 (bird infection stage). MaRNAV-1 viruses identified from *P. vivax* samples from this study are highlighted in red. Putative protozoan hosts are coloured depending on their belonging to the Alveolates (orange dark), Stramenopiles (light orange) and Euglenozoa (blue) major eukaryotic groups. Numbers indicate the branch support from 1000 bootstrap replicates. The virus tree is mid-point rooted for clarity only. (B) Eukaryotic host evolution and timescale, adapted from [38]. The two major groups Alveolates (red) and Euglenozoa (blue) are basal and their separation potentially occurred approximately two billion years ago [38].

MaRNAV-2 sequences were detected in 6 out of the 7 individual birds carrying the *Leucocytozoon*, independently of the tissue, date of sampling or bird species collected (Table S5). Interestingly, the only *Leucocytozoon* parasites free of MaRNAV-2 (sample 9585.3 collected from an Australian Magpie) may belong to a different *Leucocytozoon* species as it is phylogenetically distinct from the cluster formed by all the other *Leucocytozoon* populations in an analysis of the cytB gene (S11A Fig).

A phylogenetic analysis of the *Leucocytozoon* MaRNAV-2 amino acid sequences revealed a strong clustering of the RdRp-segment with the *P. vivax*-infecting MaRNAV-1 viruses (Fig 8). Together, these *Plasmodium* and *Leucocytozoon*-associated viruses are likely to belong to a newly-described viral clade associated with haematozoa parasites. In addition, for both segment I (S11B Fig) and segment II (S11C Fig), the viral sequence variability between samples reflects the bird species rather than the location or the date of sampling. Interestingly, the overall level of divergence is similar between the two putative segments (between 86 and 100% identical nucleotide sites).

## Discussion

Our meta-transcriptomic study of human blood samples infected with three major *Plasmodium* species revealed the presence of a highly abundant and geographically dispersed bi-segmented RNA virus in *P. vivax* that we named Matryoshka RNA virus 1 (MaRNAV-1). To the best of our knowledge this is the first documented RNA virus associated with the genus *Plasmodium*, and more broadly in parasites of the Apicomplexa subclass hematozoa. An additional investigation of complementary data sets from the SRA similarly provided strong evidence for the presence of MaRNAV-1 in *P. vivax*, but not in any of the other *Plasmodium* species

analyzed. Similarly, MaRNAV-1 was absent from *Plasmodium*-free *Anopheles* species. Notably, MaRNAV-1 is both highly conserved and at high prevalence in *P. vivax* populations from both South East Asia and South America. Finally, we documented closely related-viral sequences—MaRNAV-2—in avian samples infected with *Plasmodium*-related *Leucocytozoon* parasites. The divergent nature of the *Plasmodium* and *Leucocytozoon* viruses identified here, including the presence of a second genome segment, raises the possibility that they should be classified as a new genus or into a larger family together with LepseyNLV1.

The first segment of MaRNAV-1 encodes a single ORF containing the conserved RdRp-motif that is related to those found in the *Narnaviridae*, while the second segment, which is not characteristic of narnaviruses, encodes two overlapping ORFs of unknown function. The family *Narnaviridae* comprises a capsid-less viral family that infects plants, fungi and protists. Interestingly, no sequences associated with fungi were observed in our samples, again compatible with the idea that this virus is indeed associated with *Plasmodium*. In addition, the closest RNA virus homologs were also observed in protozoans, or in arthropods that could conceivably be infected by protozoan parasites [34, 40]. This virus-protist association evidence was reinforced by the consistent link between this virus and *P. vivax*, rather than to the metazoan hosts (mosquitoes and human) from which the samples were extracted, or the other *Plasmodium* species, including its absence in *Plasmodium*-free *Anopheles*.

The evolutionary origin of these novel *Plasmodium* and *Leucocytozoon* viruses is less clear, but can be framed as two hypotheses: (i) an ancient virus-host co-divergence between the Euglenozoa (e.g. *Leptomonas*) and Alveolate (including the hematozoa) groups of eukaryotes at almost two billion years ago [41] (Fig 9A), or (ii) horizontal virus transfer events between either a secondary (likely invertebrate) host and protozoan parasites, or among two protozoan parasites co-infecting the same secondary host, over an unknown time-scale (Fig 9B). As RNA viruses normally exhibit high rates of evolutionary change [42], the recognizable sequence similarity between the narna-like RNA viruses from *Plasmodium* (Alveolates, Apicomplexa) and *Leptomonas* (Euglenozoa, kinetoplastids) means it is perhaps unlikely that they shared a common ancestor that lived approximately two billion years ago (Figs 8B and 9). Some Euglenozoa and Alveolates independently evolved a parasitic lifestyle by infecting invertebrates and, more recently, vertebrate hosts. Hence, it is more likely that the protozoan narnavirus-like similarities reflects viral cross-species transmission between two protozoan parasites during mixed-infection in either vertebrate or invertebrate hosts (Fig 9B). The wide distribution and prevalence of MaRNAV-1 in *P. vivax* populations, as well as in the different species of *Leishmania* parasites investigated previously, supports the idea that this host jumping event is relatively ancient, although the exact time-scale is difficult to determine. As previously demonstrated, invertebrates play a key role in RNA virus evolution by feeding on many different hosts and transmitting viruses, fungi and protozoa among both plants and vertebrates [41, 43]. This may also explain why narnaviruses or closely related RNA virus have been able to spread to such a diverse range of eukaryotes, including Fungi, Stramenopiles, Alveolates and Euglenozoa. Moreover, the recent characterization of a narnavirus in the plant-infecting trypanosomatid *Phytomonas serpens* [44] suggests that vertebrates are not likely to be the hosts where the horizontal virus transfer occurred.

The RdRp segment of MaRNAV-1 documented here is clearly related to the narnavirus RdRp, although we are unable to identify a clear homolog for the second, divergent segment. Hence, as previously hypothesized for the tri-segmented plant RNA virus ourmiaviruses that combine a *Narnaviridae*-like RdRp and *Tombusviridae*-like movement and capsid proteins [45], our newly-described viruses may have evolved by reassortment of different RNA viruses during co-infection, resulting in the combination of RdRp from Narnavirus and another two ORFs from an undescribed yet RNA virus family or families (Fig 9B). Further investigation of

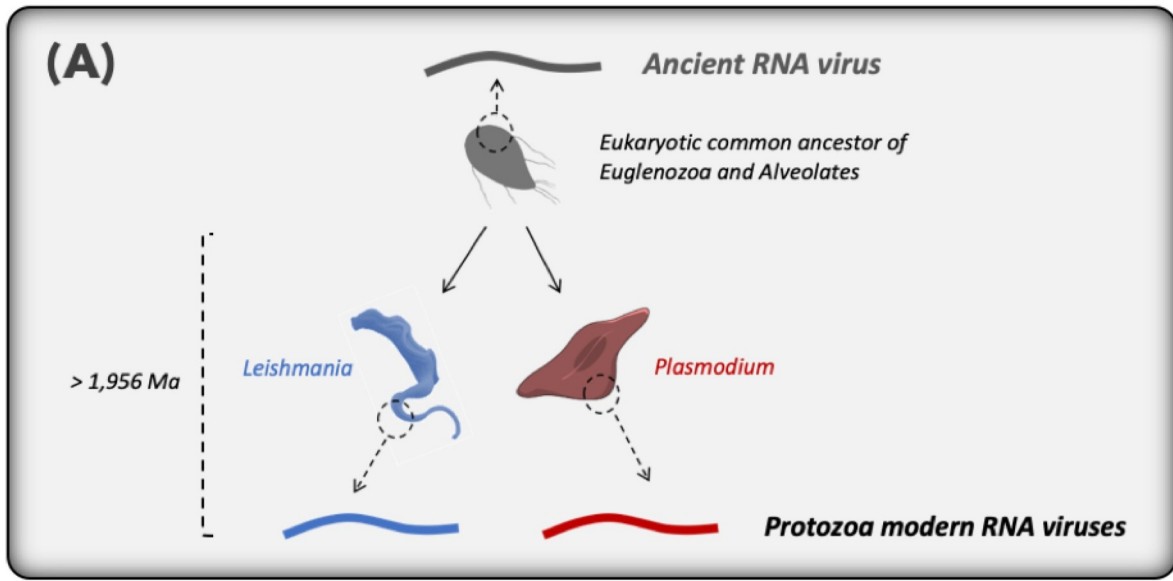

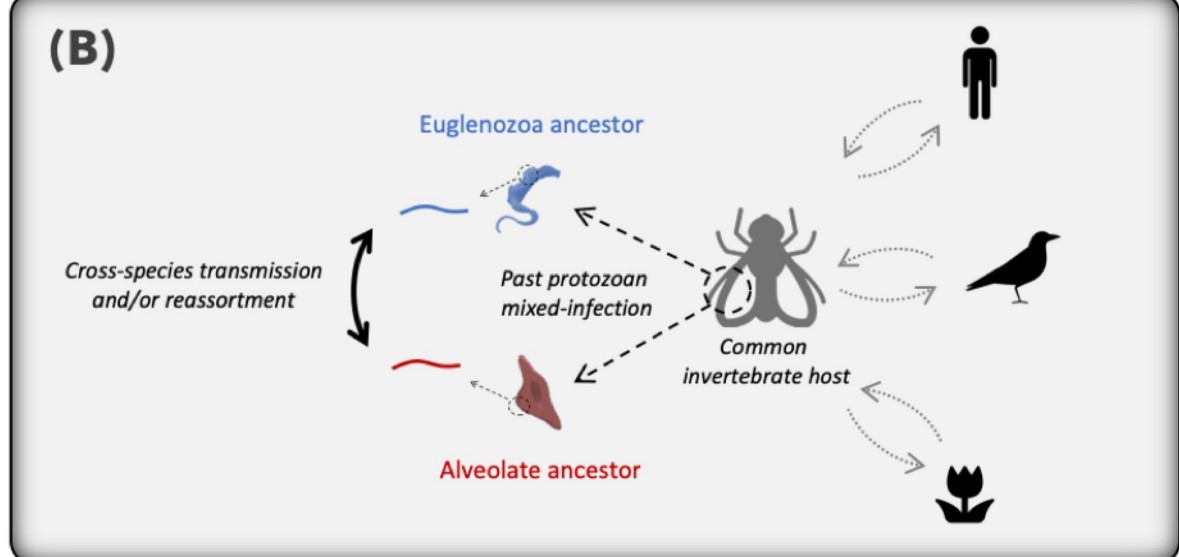

**Fig 9. Hypothetical scenarios for the origin and evolution of MaRNAV-1 and MaRNAV-2 and relatives among parasites belonging to the Alveolates and Euglenozoa eukaryotic groups.** (A) Ancient virus-host co-divergence between the Euglenozoa and Alveolates that may have occurred approximately two billion years ago. (B) Horizontal virus transfer between the Alveolates and Euglenozoa parasites that co-infected the same secondary (likely invertebrate) host.

the origins and functions of these hypothetical proteins clearly need to be conducted to better understand the virus biological cycle and its evolutionary history. Indeed, capsid-less elements cannot exist in an extracellular state and are necessarily transmitted via intracellular mode (cell division or cell mating) [46, 47].

The *Narnaviridae* comprise some of the simplest viruses described to date, containing a single segment encoding a single replicase. Despite this, they are still able to impact their hosts in profound ways. For example, a reduction in host virulence (hypovirulence) has been documented in the case of mitoviruses (a genus of *Narnaviridae* infecting fungi) [47]. Combined with the previously reported impacts of dsRNA viral infections on the biology, pathogenesis

and treatment response of the human parasite *Leishmania* [20], understanding effects of on *P. vivax* biology and pathogenesis is clearly an area of interest, although this will require confirmation of the replicative activity of the newly discovered viruses in *Plasmodium* as well as a greater understanding of the biology of infection. Intuitively, the biological consequences of the high prevalence of this virus in *P. vivax*-infected individuals will represent an important avenue for future research. More broadly, the characterization of the viral cycle of MaRNAV-1, their biology, and interactions with its host may also help to understand the biology and life-cycle of *P. vivax* parasites, as well as the modulation of host and parasite responses leading to immunoevasion, pathogenesis and transmission. Similarly, additional study of those Pv-infected samples that do not carry MaRNAV-1 may reveal some of the possible impacts of viral infection on parasite biology. Future work should also focus on the impact of virus infection on the hypnozoite liver stage of *P. vivax*, which is not present among the other, virus-negative, *Plasmodium* species assessed in this study, and is responsible for *P. vivax* infection relapses in human hosts and ongoing transmission in the absence of specific liver treatment. Finally, it will be important to determine whether MaRNAV-1 or a related virus infects the other human *Plasmodium spp.* with the hypnozoite life-cycle stage—*P. ovale curtisi* and *P. ovale wallikeri* [48]—as well as the possible role of viral infection in promoting immunoevasion, such as asymptomatic infection [49] or pathogenesis [50]. Most of the samples tested here correspond to symptomatic forms of *Plasmodium* infection, and it is possible that different virus-parasite interactions are seen in asymptomatic cases.

## Materials and methods

### Ethics statement

The study was approved by the ethics committees of the Malaysian Ministry of Health (NMRR-10-754-6684) and Menzies School of Health Research (HREC 2010–1431). All adults provided written informed consent, including the parents or guardians of children aged less than 18-years.

### Biological samples

Whole blood samples (1 ml) were collected at district hospitals from healthy and *Plasmodium*-infected patients in the state of Sabah, east Malaysia in 2013–14. Patients had a clinical illness consistent with malaria, with blood collected prior to antimalarial treatment. Parasite density was quantified by research microscopy using pre-treatment slides and reported as the number of parasites per 200 leukocytes from thick blood film. This was converted into the number of parasites per microliter using the patient's leukocyte count from their hospital automated hematology result. Remaining blood samples were stored in RNAlater and conserved at -80˚C until RNA extraction. Sampling locations, sampling dates, *Plasmodium* species validation and parasite counts are reported in S1 Table.

PCR validation for *P. vivax* and *P. falciparum* were conducted following Padley *et al.* [51]. A single-round PCR was performed using one single reverse primer in combination with species-specific forward primers (S6 Table). The *P. knowlesi*-infected sample validation were conducted following Imwong *et al.* [52] using a nested-PCR strategy with two primer couples: rPLU3 and rPLU4 for the first PCR, and PkF1140 and PkR1150 for the nested PCR (see S6 Table for the corresponding sequences).

### Total RNA extraction and RNA sequencing

Total RNAs were extracted from blood samples using the Qiagen RNeasy Plus Universal MIDI kit and following manufacturer's instructions. Importantly, randomized and serial extractions

were conducted to prevent potential experimental biases and to facilitate the detection of kit, columns, reagents or extraction-specific contamination from the corresponding meta-transcriptomic data.

Total RNA samples were grouped by *Plasmodium* species and pooled in equimolar ratios into a single sample. Quality assessments were then conducted and four TruSeq stranded libraries were synthetized by the Australian Genome Research facility (AGRF), including a rRNA and globin mRNA depletion using RiboZero and globin depletion kit from Illumina. Resulting libraries were run on HiSeq2500 (paired-end, 100bp) platform at the AGRF (RNA sample quality and the features of each library are described in S7 Table).

## rRNA and host read depletion

Raw reads were first trimmed using the Trimmomatic software [53] to remove Illumina adapters and low-quality bases. Human, rRNA and *Plasmodium* associated reads were removed from the data sets by successively mapping the trimmed reads to the latest versions of each corresponding reference sequence databases (see S8 Table for more details) with either SortmeRNA [54] or Bowtie2 software, and by applying local and very-sensitive options for the alignment [55]. All corresponding databases and the software used for the host analyses and rRNA depletions are summarized in S8 Table.

## Contig assembly and counting

Depleted read data sets were assembled into longer contigs using the Trinity software [56]. The resulting contig abundances were estimated using the RSEM software [57].

## Virus discovery

A global sequence-based homology detection was performed using BLASTn and Diamond BLASTx [58] against the entire non-redundant nucleotide (nt) or protein (nr) databases with using e-values of 1e-10 and 1e-5, respectively. Profiling plots were obtained by clustering contigs based on the taxonomy of their best BLASTn and/or BLASTx hits (highest BLAST score) and plotting their respective length and abundance using ggplot2 [59].

In parallel, a RNA virus-specific sequence-based homology detection was conducted by first aligning our data sets to a viral RdRp database using Diamond BLASTx. To ensure the removal of false-positives, a second BLASTx round using exhaustive hits output parameters was performed on each RdRp-matched contig to discard those that are more likely from a non-viral source. True-positive viral contigs were merged when possible and further analyzed using Geneious 11.1.4 software [60].

"Unknown contigs" (i.e. contigs with no BLASTx hit) longer than 1kb were retained and submitted to a second round of BLASTx using a low-stringent cut-off of 1e-4. HMM-profile and structural-based homology searches were also performed on these unknown contigs using the normal mode search of the Protein Homology/analogY Recognition Engine v 2.0 (Phyre2) web portal [36]. Briefly, the amino acid sequences of predicted ORFs were first compared to a curated non-redundant nr20 data set (comprising only sequences with <20% mutual identity) using HHblits [61]. Secondary structures were predicted from the multiple sequence alignment and this information was converted into a Hidden Markov model (HMM). This HMM was then used as a query against a HMM database built from proteins of known 3D-structures and using HHsearch [62]. Finally, a 3D-structure modelling step was performed using HHsearch hits as templates, following the method described in [61].

Virus-like sequences were further experimentally confirmed in total RNA samples by performing cDNA synthesis using the SuperScript IV reverse transcriptase (Invitrogen, Catalog

number: 11756500) and PCR amplification with virus candidate specific primers using the Platinum SuperFi DNA polymerase (Invitrogen, Catalog number: 12359010). Amplified products were Sanger sequenced using intermediary primers enabling a full-length coverage (all primers are listed in S6 Table).

### Host-virus assignment

To help assign a virus to a specific host (i.e. determining which host organism these viruses likely infect), we analyzed both codon usage bias and genomic dinucleotide composition [38, 63]. Accordingly, the average codon usage of *H. sapiens* and *P. vivax* were retrieved from the Codon Usage Database (available at http://www.kazusa.or.jp/codon/) and the codon adaptation index (CAI) and associated expected-CAI (eCAI) were determined using the CAIcal webserver, available at http://genomes.urv.es/CAIcal/E-CAI/ [64]. As the most prevalent *Anopheles* mosquito vector in the Sabah region of Malaysia (*Anopheles balabacensis*) did not have a codon usage table available, in this case we retrieved the codon usage from seven other *Anopheles* species (*A. dirus*, *A. minimus*, *A. cracens*, *A. gambiae*, *A. culicifacies*, *A. merus* and *A. stephensi*) which included major vectors of malaria in South East Asia.

As well as codon bias, we determined the dinucleotide composition of MaRNAV-1 and compared to that of *Anopheles gambiae* (RefSeq | GCF_000005575.2) for which a full-genome sequence is available, and *P. vivax* (RefSeq | GCF_000002415.2). The match between host and virus was then calculated using the method described previously [38] by calculating the $f_{xy}/f_xf_y$ ratio from the MaRNAV-1 sequences obtained by Sanger sequencing (see above).

### Virus genome characterization

Validated virus-like genomes were further characterized using both genome/protein annotation programs, including InterProScan for protein domain, Sigcleave and Fuzzpro from EMBOSS package for signal cleavage sites and motifs, and TMHMM for transmembrane regions [65, 66].

### Mining of the Sequence Read Archive (SRA)

To identify homologs of MaRNAV-1, the newly identified Narna-like virus sequence was used as a reference in both Magic-BLAST BLASTn (default parameters) [67] and Diamond BLASTx (cut-off 1e-5) [58] analyses of several RNA-seq data sets obtained from *Plasmodium sp*. available on the NCBI SRA using the NCBI SRA toolkit v2.9.2. The list of the corresponding BioProjects, runs and references are provided in S2 Table.

*P. vivax* SRA library information (i.e. host, location and biological replicates) was manually retrieved from the corresponding papers (S3 Table). Where possible, this information was used to assess whether such variables were associated with the detection of narna-like viruses by performing Chi-squared tests using the SPSS Statistics software (IBM). SRA runs positive for homologs to MaRNAV-1 (number of read BLAST hits >100) were imported locally and assembled following the same workflow as previously used to extract homologous full-length contigs and to calculate their relative abundance in the samples.

To further assess host assignments, the same SRA data sets were subjected to Magic-BLAST using *Plasmodium* and mosquito and human specific housekeeping gene transcripts (LDH-P gb|M93720.1 and RSP-7 gb|L20837.1, respectively). This large-scale analysis may necessarily result in false-negative results because of the idiosyncrasies of the experimental procedures used, such as the depletion of human reads or *Plasmodium* RNA enrichment, both of which can bias such host read counting. Moreover, some samples come from the same biological replicate and hence cannot be counted as independent. Such potential biases

forced us to manually retrieve all information for each *P. vivax* SRA library using related publications (S3 Table).

## Phylogenetic analysis

Predicted ORFs containing the viral RNA-dependent (RdRp) domain from both the SRA and human blood and bird associated sequences (see below) were translated and aligned using the E-INS-I algorithm in MAFFT v7.309 [68]. To place the newly identified viruses into a more expansive phylogeny of RNA viruses, reference protein sequences of the closest homologous viral families or genera were retrieved from NCBI and incorporated to the amino acid sequence alignment. The alignments were then trimmed with Gblocks under the lowest stringency parameters [69]. The best-fit amino acid substitution models were then inferred from each curated protein alignment using either the Smart model selection (SMS) [70] or Model-Finder [71], and maximum likelihood phylogenetic trees were then estimated with either PhyML [72] or IQ-tree [73] with bootstrapping (1000 replicates) used to assess node support. For clarity, all phylogenetic trees were midpoint rooted.

## Analysis of avian meta-transcriptomic libraries

To supplement our analysis of human *Plasmodium* samples, we analyzed four meta-transcriptomic libraries sampled from four bird species (*Gymnorhina tibicen*, *Strepera graculina*, *Corvus coronoides* and *Grallina cyanoleuca*) in New South Wales, Australia. Sampling details are reported in S5 Table. The RNA-Seq data analysis and the BLASTx detection of MaRNAV-1 homologs from bird sample data sets were conducted as described above.

The PCR-based detection of both narna-like viruses and *Leucocytozoon* parasites were conducted using newly-designed primers corresponding to the *Leucocytozoon* homologs of the MaRNAV-1 RdRp and segment II (primers are described in S6 Table), and following the same PCR protocol as described above. PCR-based *Leucocytozoon* detections were performed using primers targeting the *Leucocytozoon* mitochondrial cytochrome B oxidase gene as described in [74]. All additional analyses of the bird data sets were performed utilizing the software and tools described above.

## Supporting information

**S1 Table. Description of human blood samples used in this study.** PCR: PCR-based validation of *Plasmodium* species using species-specific primers: Pv—*P. vivax*; Pk—*P. knowlesi*; Pf—*P. falciparum*; pc—parasite counting (i.e. parasite density / μL = number of parasites counted x patient's lab leukocyte result / 200 leukocytes counted).
(DOCX)

**S2 Table. BioProject and corresponding SRA accessions used in this study.**
(DOCX)

**S3 Table. *Plasmodium vivax* SRA libraries.** Libraries positive for MaRNAV-1 are shown in red and those used for phylogenetic analysis (MaRNAV-1-like read counts > 100) are shown in bold. *Plasmodium*-free libraries are in grey. *Homo sapiens* = human blood samples. *Anopheles dirus* = mosquito salivary gland dissection samples. *Homo sapiens (ex vivo)*: Micropatterned cellular co-cultures. ND: Not determined.
(DOCX)

**S4 Table. BioProject and corresponding SRA accessions of *P. vivax*-free Anopheles species.**
(DOCX)

**S5 Table. Bird sample analysis summary table.** Presence, absence or uncertainty of *Leucocytozoon* detection from pathology reports are highlighted in green, red and orange, respectively. Positive or negative PCR targeting either the *Leucocytozoon* parasite, the RdRP-like segment and the unknown second segment of MaRNAV-2 are highlighted in green and red, respectively.
(DOCX)

**S6 Table. Primers used in this study.**
(DOCX)

**S7 Table. Quality of RNA extraction and RNA-seq data sets obtained.**
(DOCX)

**S8 Table. List of databases and software used for rRNA and host read depletion.**
(DOCX)

**S1 Fig. *Plasmodium* parasite count in human blood samples.** Parasite counts are expressed as the number of parasites per μl of blood.
(TIF)

**S2 Fig. Results of the Trinity sequence assembly.** Contig length and count obtained after performing Trinity assembly of libraries depleted in rRNA, human and *Plasmodium* reads.
(TIF)

**S3 Fig. Orphan contig length and abundance.** An arbitrary cut-off of 1000 nt was used to identify candidate RNA viruses. Abundance is expressed using the expected count value provided by the RSEM analysis.
(TIF)

**S4 Fig. Length and abundance of contigs from candidate non-major host taxa (BLASTn/ BLASTx).** Abundance is expressed using the expected count value provided by the RSEM analysis.
(TIF)

**S5 Fig. Comparison of CAI values obtained by comparing codon usage of MaRNAV-1 viral contigs retrieved from each *Plasmodium* library to the potential hosts *P. vivax, H. sapiens* and mosquitoes of the genus *Anopheles*.**
(TIF)

**S6 Fig. Odds ratios of dinucleotides (fxy/fxfy) from MaRNAV-1 contigs (bottom) versus *P. vivax* (top, right) and *A. gambiae* (top, left) genomic sequence.**
(TIF)

**S7 Fig. Association study between MaRNAV-1 contigs and the characteristics of *P. vivax* libraries at the SRA (Chi-squared tests).** (A) *Plasmodium* infection association test. (B) Biological replicates association test. Replicates corresponding to the same biological sample are grouped by colour. (C) Sample location association test. (D) Host association test.
(TIF)

**S8 Fig. Read count mapping to the MaRNAV-1 segment I (x-axis, log scaled) and Segment II (y-axis, log scaled) in *P. vivax* SRA data sets.** The R-squared value is indicated on the right.
(TIF)

**S9 Fig. Sequence alignment (MAFFT) of MaRNAV-2 contigs in *Leucocytozoon*-infected bird and MaRNAV-1 sequences in *P. vivax*-infected humans.** (A) Analysis of segment I. (B)

Analysis of segment II. Nucleotide alignments are shown on the left and protein alignments are shown on the right (top—ORF1; bottom—ORF2). Orange boxes: predicted ORF using standard genetic code. Light green boxes: InterProScan domain prediction. Yellow to green plots: level of sequence conservation.
(TIF)

**S10 Fig. PCR-based detection of leucocytozoons and MaRNAV-2 (2 segments) in avian cDNA samples.** Top: *Leucocytozoon* CytB PCR; Middle: MaRNAV-2 segment I homolog detection; Bottom: MaRNAV-2 segment II homolog detection.
(TIF)

**S11 Fig. Maximum likelihood phylogenetic trees of parasites and MaRNAV-2 homologs obtained from bird samples.** (A) Hematozoa CytB phylogeny. The CytB sequence from *Gallus gallus* is used as an outgroup. Samples positive for MaRNAV-2 l are marked with *. (B) MaRNAV-2 segment I phylogeny; (C) MaRNAV-2 segment II phylogeny. Sequences from bird samples are shown in red.
(TIF)

**S1 References. Supporting references.**
(DOCX)

## Acknowledgments

We acknowledge the Sydney Informatics Hub and the University of Sydney's high performance cluster Artemis for providing the computational resources required for the RNA-Seq data processing, and Wei-Shan Chang for providing technical assistance with PCR validation. We thank Dr. Andrew Harmon and Heeva Baharlou (Westmead Institute for Medical Research) for valuable input. We also thank the Director-General, Ministry of Health, Malaysia, for permission to publish this manuscript.

## Author Contributions

**Conceptualization:** Justine Charon, Matthew J. Grigg, John-Sebastian Eden, Miles P. Davenport, Nicholas M. Anstey, Edward C. Holmes.

**Data curation:** Justine Charon, John-Sebastian Eden.

**Formal analysis:** Justine Charon, Hafsa Rana, Edward C. Holmes.

**Funding acquisition:** Matthew J. Grigg, Nicholas M. Anstey, Edward C. Holmes.

**Investigation:** Justine Charon, Hafsa Rana, Karrie Rose.

**Methodology:** Justine Charon, Matthew J. Grigg, John-Sebastian Eden, Kim A. Piera, Timothy William, Karrie Rose, Miles P. Davenport, Nicholas M. Anstey, Edward C. Holmes.

**Resources:** Matthew J. Grigg, John-Sebastian Eden, Kim A. Piera, Timothy William, Karrie Rose.

**Supervision:** Edward C. Holmes.

**Validation:** Justine Charon, Matthew J. Grigg, John-Sebastian Eden, Edward C. Holmes.

**Visualization:** Justine Charon, Edward C. Holmes.

**Writing – original draft:** Justine Charon.

**Writing – review & editing:** Matthew J. Grigg, John-Sebastian Eden, Kim A. Piera, Hafsa Rana, Miles P. Davenport, Nicholas M. Anstey, Edward C. Holmes.

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
