## [Decision Letter · Decision Letter 0]

2 Aug 2019

Dear Dr. Holmes:

Thank you very much for submitting your manuscript "Novel RNA viruses associated with Plasmodium vivax in human malaria and Leucocytozoon parasites in avian disease" (PPATHOGENS-D-19-00967) for review by PLOS Pathogens. Your manuscript was fully evaluated at the editorial level and by two independent peer reviewers. The reviewers appreciated the attention to an important topic but identified some aspects of the manuscript that should be improved.

While both reviewers consider your discovery of an RNA virus in Plasmodium vivax and Leucocytozoon parasites important and well supported by the data, reviewer 1 points to the exclusively sequencing-based nature of the evidence and would like to see additional experimental work that the viral polymerase or RNA is actually present within the parasite. At one level, we agree with the reviewer and think that such additional data would add a very relevant new dimension to your work by allowing the new virus to be localised unequivocally within the parasite. We ask you to consider adding such data, in particular since the in situ hybridisation approach using RNAscope reagents should be highly sensitive and could be executed within a matter of weeks. On the other hand we recognise that some of the work suggested by reviewer 1 would take considerably more time and we would rather like to see you deal with some of their concerns in the discussion, rather than delay resubmission of your manuscript substantially, given that both reviewers consider your major conclusions broadly supported by different lines of evidence.  

In any case, we ask you to modify the manuscript according to the review recommendations before we can consider your manuscript for acceptance. Your revisions should address the specific points made by each reviewer.

(1) A letter containing a detailed list of your responses to the review comments and a description of the changes you have made in the manuscript. Please note while forming your response, if your article is accepted, you may have the opportunity to make the peer review history publicly available. The record will include editor decision letters (with reviews) and your responses to reviewer comments. If eligible, we will contact you to opt in or out.

(2) Two versions of the manuscript: one with either highlights or tracked changes denoting where the text has been changed; the other a clean version (uploaded as the manuscript file).

We hope to receive your revised manuscript within 60 days or less. If you anticipate any delay in its return, we ask that you let us know the expected resubmission date by replying to this email.

[LINK]

Sincerely,

Oliver Billker

Associate Editor

PLOS Pathogens

Raul Andino

Section Editor

PLOS Pathogens

Kasturi Haldar

Editor-in-Chief

PLOS Pathogens

orcid.org/0000-0001-5065-158X

Grant McFadden

Editor-in-Chief

PLOS Pathogens

orcid.org/0000-0002-2556-3526

Reviewer's Responses to Questions

**Part I - Summary**

Reviewer #1: This manuscript describes the detection of a narnavirus-like sequence in RNA-seq data obtained from Plasmodium vivax-infected individuals. This virus is found at a relatively high viral load, has a friendly segment at a similar copy number, and is only associated with P. vivax infection. Narnaviruses have been found in protozoa before, but there's been no virus associated with plasmodium - narnaviruses have typically been considered lame viruses in the world of viral discovery since they were thought to have no structural proteins, but that may require reconsidering with the discoveries of second segments associated with these viruses. The use of SRA data to better link host and virus is a strong addition to the work and very helpful.

Reviewer #2: This manuscript reports the identification of MaRNAV-1, a narnavirus-like bisegmented sequence that encodes a conserved RNA polymerase in its first segment, and two unidentified ORFs in its second segment, in the human malaria parasite Plasmodium vivax. The authors used a meta-transcriptomic study of blood samples from patients with P. vivax, P. falciparum and P. knowlesi, and confirmed their finding through mining of RNA-seq datasets P. chabaudi, P. cynomolgi, P. falciparum, P. yoelli, P. knowlesi and P. berghei available in public databases. Additionally, reference assisted alignments using the MaRNAV-1 sequences on Australian birds infected with Leucocytozoon, discovered MaRNAV-2. This parasite is a distant relative of Plasmodium but it belongs to the same Apicomplexa subclass hematozoa.

These are the first two RNA viruses reported to infect hematozoa and thus this manuscript represents a highly interesting and novel finding. It also broadens our understanding of viral evolution in eukaryotic microbes, while revealing the possibility of host-pathogen-viral interactions that could be important for our understanding of the P. vivax parasite. However, there are several instances of missing data and text, and overall the paper's claims and description could be made a lot tighter.

**Part II – Major Issues: Key Experiments Required for Acceptance**

Reviewer #1: The data presented here are very intriguing and the authors are most likely correct, but my enthusiasm is tempered by the lack of biological experiments to 1) conclusively associate the host-virus relationship, 2) test for presence of nucleic acid or protein in the parasites themselves, or 3) to extend understanding of the second segment. It’s not just the revenge of the wet-lab folk; it’s important when doing less controlled — dare we say, descriptive — studies of human infections, as other viral discoveries have this problem, especially in vector-borne diseases where the vector may be carrying multiple viruses or protozoa and/or they may be in areas where multiple vectors exist. There are opportunities to go beyond the initial detection of the virus described here in this report.

Probably most germane is #2 above. Can the authors raise antibody to the viral polymerase and show it is present within the parasite and/or perform RNAscope and show that the segments are generally within the parasite?

Also worthwhile would be to conclusively show that the virus is not present in different species of Anopheles mosquitos when P. vivax is not present, as that is the most likely confounder for these types of metagenomic studies in human specimens who have been bitten by so many vectors (again the SRA data is nicely done here, but it is not clear that it is adequately sampling different Anopheles vectors).

Farther along the line, despite the difficulty of P. vivax culture, it would be important to see if the virus can grow with P. vivax in vitro for a handful of cycles; or one could obtain material from controlled human infections with vivax to see if the virus persists.

Reviewer #2: (No Response)

**Part III – Minor Issues: Editorial and Data Presentation Modifications**

Reviewer #1: line 334 - I think given the bioinformatic and RNA detection focus of this paper, there's an opportunity to get more hints at the biology from some analysis of the P. vivax libraries that did NOT have MaRNA-1 in them.

line 159 - would put a unit on the parasitemia numbers.

line 421 - would mention LepseyNLV1 here since it also has two segments and is narna-like. One would expect that if you

line 443 - it is not clear that the same rapid evolutionary rates hold for narnaviruses.

Figure 2 - pity to have all that computational knowledge and firepower, but can't get an agarose gel to run straight or rotate an image...

Figure 3 - I did not see the authors conjectures as to why there is so much divergence in Pv#2 in RdRp compared to other viruses, and why there is more divergence in the RdRp compared to the second segment.

Figure 4 - I did not see a mention of the reads to MaRNA-1 in P. cynomolgi or other Plasmodium in the text...

Figure 7 - unclear labels in the legend.

Reviewer #2: Main comments

1. Table 2 is missing from the manuscript, referred to in lines 204 and 221. Apparently it contains important information on the RNA virus-like contigs, such as their length and abundance (RPKM), but this information was not available for review.

2. Figure 2a, Plasmodium falciparum gel image: This figure appears to be composed of 3 different gels, one for each of the amplicons. The left side of the gel for the Human RPS18 lacks a MW ladder, which makes the sizes of the amplicons unclear. So we are unable to confirm the validity of this figure.

3. Does P. vivax infect Anopheles gambiae? It's unclear what the relevance of undertaking dinucleotide composition analysis of this species of mosquito is, if it's not routinely a vector of Pv

Minor comments

1. Figure 1A is confusing, and requires additional information to explain the filtering of the dataset. Lines 165 and 166 mention that the number of reads filtered from the non-infected, Pf, Pk, and Pv data sets were 93%, 81%, 42%, and 57% respectively. Figure 1A shows bar graphs that represent the efficiency of read filtering, or the number of reads remaining once a specific type of sequence is filtered (i.e., w/o Human, w/o rRNA, and w/o Plasmodium). Shouldn't the percentage of reads filtered (reported on Lines 165 & 166) be equal to the combined proportion of the number of reads filtered for all the filtering criteria in Figure 1A? Based on read numbers taken from Figure 1A, we can calculate the total percentage of reads kept after filtering for each of the individual conditions (w/o Human, w/o rRNA, or w/o Plasmodium). For example, Pv read numbers after filtering on Figure 1A show that the percentage of reads left after individual filtering are as follows: w/o Human = 43%, w/o rRNA = 35.49 %, and w/o Plasmodium = 4.5%. If we subtract the percentage of the category with the most reads filtered from the raw percentage of reads (100%), we get the value reported for the percentage number of filtered reads reported on lines 165 & 166. For Pv this calculation would be (100% Raw Pv reads)-(43% Pv reads left w/o Human)=(57% reads filtered for Pv). However, this calculation for total proportion of filtered reads does not account for cases in which the dataset is being additionally filtered using the other two filtering criteria with lower percentages (in the case of Pv: w/o rRNA = 35.99% and w/o Plasmodium= 4.5%). Given that all libraries on Figure 1A follow the same patterns of percentages of reads left after filtering and filtering criteria (i.e. w/o Human = (highest percentage), w/o rRNA = (percentage somewhere between Highest and lowest), and w/o Plasmodium = (lowest percentage)), it suggests that filtering is occurring sequentially and not in parallel for each of the filtering criteria. If this is the case, the figure legend should be changed perhaps to the following:

a. (black) Raw reads

b. (grey) Trimmed reads

c. (blue) w/o human.

d. (yellow) w/o Human and w/o rRNA

e. (orange) w/o Human, w/o rRNA, and w/o Plasmodium.

2. Line 37: “...patients suffering malaria…” change to “...patients suffering from malaria…”.

3. Line 65: “This is first discovery…” change to “This is the first discovery....”

4. Line 155 & 156: “An additional library of 6 non-infected patients were also included…”. should be “was”. i.e.“An additional library of 6 non-infected patients was also included…

5. Line 179: “...the nr database…” define “nr”.

6. Line 184: “...Trinity assembles…”. should be plural noun “assemblies”

7. Lines 237, 238, and 239: sample numbers 2 and 10 are inconsistently used with a # (number sign) preceding the number

8. Multiple instances: “Blast” should be captalized BLAST as it is an abbreviation of Basic Local Alignment Search Tool.

9. Lines 203 and 204: “Assuming that RNA virus-like contigs would be of a certain minimum length, only those larger than 1000 nt were used for further analysis (Table 2).” Is there a ref for this?. The only reference mentioned in the current version of the text is Table 2, and this table is missing from the manuscript.

10. Line 383: “RT-PCR” is this “Real Time” or “Reverse Transcriptase” PCR?

11. Line 887: “RT-PCR of each samples...” change to “RT-PCR of each of the samples...”.

12. Figure 2A:

a. Amplicon sizes should be shown for each of the expected products.

b. Ladder size should be provided.

c. Sample numbering does not follow a regular format, would improve readability.

d. Figure 2B: Can human and Plasmodium controls be run along with the MaRNAV-1 segment II amplicons? Will provide for better comparison.

13. Figure 3: This figure isn't very informative - it is a schematic and not a multiple sequence alignment, since the sequence is not shown.

a. White text inside boxes is difficult to read.

b. Green bar at top of Fig 3b - what is this?

c. Frames in ORF1 are different between samples. These differences are not discussed, and thus their relevance is unknown

d. (Lines to 234 to 247)

i. The authors should elaborate on intra-sample polymorphism, explain what it is and how it was determined for this figure. Does intra-sample polymorphism refer to differences between Sanger-sequenced sample replicates? If so, could this be a sequencing artifact?

ii. Similarly with inter-sample polymorphism, explain what it is and how it was determined for this figure. Figure 3A (left) has an alignment of nucleotides, for ORF1 across the different Pv samples. However, it lacks a matrix with percentage of identity at the nucleotide level. This kind of matrix was included for the analysis of segment II on Figure 3B, but it should also be included for segment I on Figure 3A.

14. Figure 3B: Legend for the top and bottom parts of the figure are reversed

15. Figure 7: Legend and description of the figure needs to be expanded.

PLOS authors have the option to publish the peer review history of their article (what does this mean?). If published, this will include your full peer review and any attached files.

Reviewer #1: No

Reviewer #2: No

---

## [Editor Report · Decision Letter 1]

13 Nov 2019

Dear Dr. Holmes,

We are pleased to inform that your manuscript, "Novel RNA viruses associated with Plasmodium vivax in human malaria and Leucocytozoon parasites in avian disease", has been editorially accepted for publication at PLOS Pathogens. 

Before your manuscript can be formally accepted and sent to production, you will need to complete our formatting changes, which you will receive by email within a week. Please note that your manuscript will not be scheduled for publication until you have made the required changes.

IMPORTANT NOTES

(1) Please note, once your paper is accepted, an uncorrected proof of your manuscript will be published online ahead of the final version, unless you’ve already opted out via the online submission form. If, for any reason, you do not want an earlier version of your manuscript published online or are unsure if you have already indicated as such, please let the journal staff know immediately at plospathogens@plos.org.

(2) Copyediting and Proofreading: The corresponding author will receive a typeset proof for review, to ensure errors have not been introduced during production. Please review the PDF proof of your manuscript carefully, as this is the last chance to correct any errors. Please note that major changes, or those which affect the scientific understanding of the work, will likely cause delays to the publication date of your manuscript. 

(3) Appropriate Figure Files: Please remove all name and figure # text from your figure files. Please also take this time to check that your figures are of high resolution, which will improve the readbility of your figures and help expedite your manuscript's publication. Please note that figures must have been originally created at 300dpi or higher. Do not manually increase the resolution of your files. For instructions on how to properly obtain high quality images, please review our Figure Guidelines, with examples at: http://journals.plos.org/plospathogens/s/figures.

(4) Striking Image: Please upload a striking still image to accompany your article if one is available (you can include a new image or an existing one from within your manuscript). Should your paper be accepted, this image will be considered for our monthly issue image and may also appear on our website to feature your article. Please upload this as a separate file, selecting "striking image" as the file type upon upload. Please also include a separate "Other" file with a caption, including credits and any potential copyright information. Please do not include the caption in the main article file. If your image is from someone other than yourself, please ensure that the artist has read and agreed to the terms and conditions of the Creative Commons Attribution License at http://journals.plos.org/plospathogens/s/content-license. Please note that PLOS cannot publish copyrighted images.

(5) Press Release or Related Media: If your institution or institutions have a press office, please notify them about your upcoming paper at this point, to enable them to help maximize its impact. If they will be preparing press materials for this manuscript, please inform our press team in advance at plospathogens@plos.org as soon as possible. We ask that you contact us within one week to plan ahead of our fast Production schedule. If you need to know your paper's publication date for related media purposes, you must coordinate with our press team, and your manuscript will remain under a strict press embargo until the publication date and time. This means an early version of your manuscript will not be published ahead of your final version. 

(6)  PLOS requires an ORCID iD for all corresponding authors on papers submitted after December 6th, 2016. Please ensure that you have an ORCID iD and that it is validated in Editorial Manager.  To do this, go to ‘Update my Information’ (in the upper left-hand corner of the main menu), and click on the Fetch/Validate link next to the ORCID field.  This will take you to the ORCID site and allow you to create a new iD or authenticate a pre-existing iD in Editorial Manager

(7) Update your Profile Information: Now that your manuscript has been provisionally accepted, please log into Editorial Manager and update your profile, if needed. Go to https://www.editorialmanager.com/ppathogens, log in, and click on the "Update My Information" link at the top of the page. Please update your user information to ensure an efficient production and billing process. 

(8) LaTeX users only: Our staff will ask you to upload a TEX file in addition to the PDF before the paper can be sent to typesetting, so please carefully review our Latex Guidelines http://journals.plos.org/plospathogens/s/latex in the meantime.

(9) If you have associated protocols in protocols.io, please ensure that you make them public before publication to guarantee immediate access to the methodological details.

Best regards,

Oliver Billker

Associate Editor

PLOS Pathogens

Raul Andino

Section Editor

PLOS Pathogens

Kasturi Haldar

Editor-in-Chief

PLOS Pathogens

orcid.org/0000-0001-5065-158X

Grant McFadden

Editor-in-Chief

PLOS Pathogens

orcid.org/0000-0002-2556-3526
---

## [Editor Report · Acceptance letter]

19 Dec 2019

Dear Dr. Holmes,

We are delighted to inform you that your manuscript, "Novel RNA viruses associated with Plasmodium vivax in human malaria and Leucocytozoon parasites in avian disease," has been formally accepted for publication in PLOS Pathogens.

Best regards,

Kasturi Haldar

Editor-in-Chief

PLOS Pathogens

orcid.org/0000-0001-5065-158X

Grant McFadden

Editor-in-Chief

PLOS Pathogens

orcid.org/0000-0002-2556-3526